# From Feasible to Practical: Pareto-Optimal Synthesis Planning

**Friedrich Hastedt** [1]  **Dongda Zhang** [2]  **Antonio del Rio Chanona** [1]

## Abstract

Current computer-aided synthesis planning (CASP) methods often treat retrosynthesis as solved once a single feasible route is identified, focusing primarily on convergence or shortest-path metrics. This view is misaligned with real-world practice, where chemists must balance competing objectives such as cost, sustainability, toxicity, and overall yield. To address this, we formulate synthesis planning as a multi-objective search problem and introduce MORetro*, an algorithm that generates a Pareto front of synthesis routes to explicitly capture trade-offs among user-defined criteria. MORetro* uses weighted scalarization and BO-informed sampling to efficiently navigate the combinatorial search space and prioritize promising trade-offs. Building on multi-objective A*-search, we provide optimality guarantees showing that, for a fixed single-step model, MORetro* recovers the true Pareto front under admissibility. Across multiple retrosynthesis benchmarks, MORetro* produces diverse, high-quality Pareto fronts, uncovering solutions overlooked by single-objective approaches and better aligning CASP outputs with industrial decision-making.

## 1. Introduction

Retrosynthesis is a chemical discipline central to accelerating drug and material discovery. Starting with a target molecule, the aim of retrosynthesis is to find synthesis routes to commercially available building blocks in a backward fashion. Once a viable synthesis route is identified, the building blocks can be purchased and the physical synthesis is carried out through wet lab experimentation. However, finding such a viable synthesis route is not trivial, as each intermediate molecule in the synthesis tree could theoretically be synthesized from hundreds of chemical transformations. This becomes particularly challenging when one tries to find a synthesis route that also optimizes key metrics, such as economic and sustainability costs.

To facilitate synthesis planning, researchers proposed computational workflows that can navigate these intrinsically large search spaces efficiently (Genheden et al., 2020; Tu et al., 2025). These workflows are commonly composed of two interacting tools: i) a single-step model and ii) a multi-step planning algorithm. The single-step model is usually a black-box machine learning (ML) model that takes the product molecule as input and returns a set of possible reactants (Segler & Waller, 2017). The single-step model is optimized offline to faithfully reproduce feasible reaction chemistry. As the name suggests, the single-step model only considers one reaction at a time. However, synthesis routes usually consist of multiple sequential reactions. To ensure convergence to commercial building blocks across the immense reaction space, the multi-step algorithm iteratively calls the single-step model, thereby constructing a synthesis route (Segler & Waller, 2018). Most multi-step algorithms to date were proposed with this exact goal in mind: returning at least one synthesis route that ends in commercial building blocks. From the perspective of the search problem, it is paramount to find at least one synthesis route per target. In practice, chemists would nonetheless investigate diverse synthesis pathways to identify the ones that best meet personal objectives, which are often a combination of financial and sustainability interests.

Embedding multiple objectives directly in the search algorithm is not trivial, as it could interfere with the algorithm's ability to find synthesis pathways. That is why the multi-step algorithm is commonly used to identify an unranked set of synthesis routes. Post-hoc analysis then optimizes with specific objectives in mind (Fromer & Coley, 2024), often adding relevant conditions (temperature and agents) in the process (Sun et al., 2025). By keeping these two stages separate, it is highly probable that the algorithm misses out on promising synthesis routes.

In this paper, we address this challenge by introducing MORetro*, a search algorithm for multi-objective synthesis planning. Given an arbitrary number of user-defined objectives, MORetro* returns the Pareto front (PF), where

---

[1]Department of Chemical Engineering, Imperial College London, UK [2]Department of Chemical Engineering, University of Manchester, UK. Correspondence to: Antonio del Rio Chanona <a.del-rio-chanona@imperial.ac.uk>.

*Proceedings of the 43rd International Conference on Machine Learning*, Seoul, South Korea. PMLR 306, 2026. Copyright 2026 by the author(s).

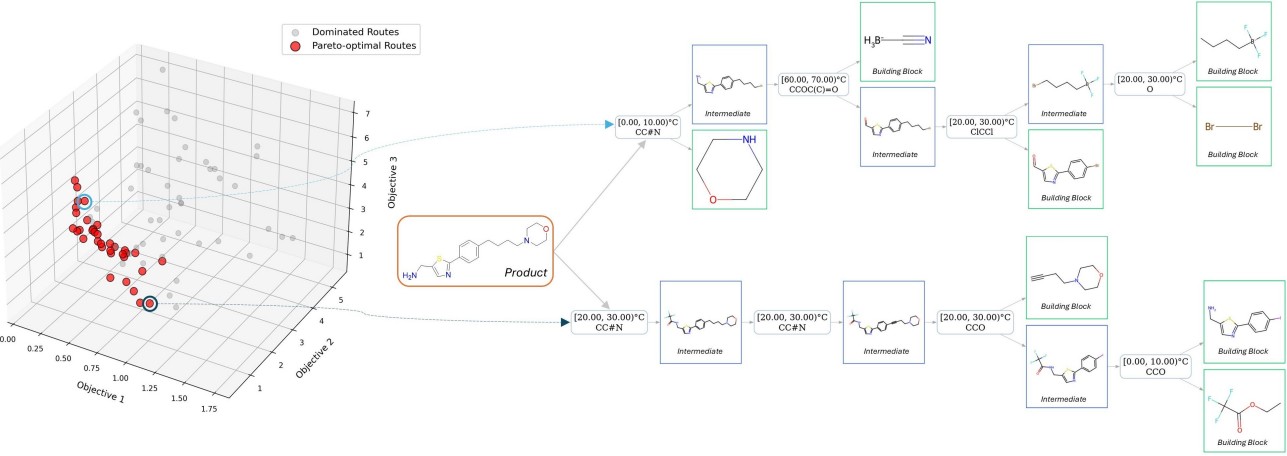

*Figure 1.* Exemplary Pareto front generated by MORetro*. Each red point corresponds to a Pareto-optimal synthesis route given an arbitrary number of user-specified objectives.

each point corresponds to a unique synthesis route (Figure 1). MORetro* extends the single-objective Retro* algorithm (Chen et al., 2020) to the multi-objective setting by down-sampling the high-dimensional objective space via linear (weight-based) scalarization. To effectively explore this space, we introduce several weight-sampling strategies. Moreover, we derive a lower bound on the multi-dimensional objective space that guarantees the returned front to be Pareto-optimal. This bound also enables early pruning of dominated intermediate molecules; those that cannot be part of any Pareto-optimal synthesis route if further expanded. Finally, MORetro* directly integrates a reaction condition prediction module, which augments each reaction with predicted conditions such as temperature and reagents during the search. This integration allows us to define and optimize novel objectives of practical relevance to the chemical industry, including sustainability and scalability.

Our contributions can be summarized as follows:

- We introduce a multi-objective retrosynthesis search algorithm that returns the Pareto front for any number of user-defined objectives, using weight scalarization and sampling. It outperforms single-objective baselines in both Pareto front quality and diversity, independent of the single-step model.

- We derive optimality bounds guaranteeing that, for a given single-step model, our algorithm can recover all Pareto-optimal synthesis routes.

- We define industrially relevant search objectives capturing toxicity, sustainability, and scale-up potential, enabling the algorithm to prioritize practically feasible reactions.

## 2. Background

### 2.1. Related Work

**Computer-aided Synthesis Planning** Retrosynthetic planning is commonly formulated as a tree-search problem, where branches correspond to chemical transformations predicted by a single-step model. These transformations are often represented as reaction templates, leading to *template-based* approaches that rely on neural policies (Dai et al., 2019; Chen & Jung, 2021) or expert rules (Szymkuć et al., 2016). In contrast, *template-free* methods avoid predefined rules and formulate retrosynthesis as sequence-to-sequence translation (*e.g.*, SMILES) (Schwaller et al., 2019; Karpov et al., 2019) or molecular graph editing (Sacha et al., 2021; Somnath et al., 2021; Zhong et al., 2023). Extending single-step prediction to multi-step synthesis requires a search algorithm to select promising intermediates during tree expansion. Monte Carlo Tree Search (MCTS) addresses this via rollout-based exploration-exploitation trade-offs (Segler & Waller, 2018). Alternatives avoid rollout by using AND-OR trees with heuristic guidance, either hand-crafted or neural (Kishimoto et al., 2019; Chen et al., 2020), with Chen et al. (2020) proposing a neural-guided A*-like algorithm. Subsequent work builds on MCTS and A* to improve route-finding success rates (Kim et al., 2021; Xie et al., 2022; Tripp et al., 2024; Zhao et al., 2024). A parallel line of work applies reinforcement learning to fine-tune single-step models in the multi-step setting (Schreck et al., 2019; Yu et al., 2022; Liu et al., 2023; Wang & Montana, 2025). Unlike our approach, these methods primarily aim to maximize the likelihood of discovering *any* valid synthesis route.

**Constrained Synthesis Planning** Recent studies incorporate explicit constraints into retrosynthesis search, which, while not framed as multi-objective optimization, are conceptually related, as constraints impose simultaneous re-

quirements on valid routes. Examples include starting-material-constrained retrosynthesis targeting specific building blocks (Yu et al., 2024; Armstrong et al., 2025), bond constraints that restrict which bonds may be broken or preserved (Thakkar et al., 2023; Kreutter & Reymond, 2023; Westerlund et al., 2025), and approaches that condition retrosynthesis on human prompts using large language models (Xuan-Vu et al., 2025).

**Multi-objective Synthesis Planning** Lai et al. (2025) first explored multi-objective synthesis planning, proposing a multi-objective Monte Carlo Tree Search (MO-MCTS) framework that maintains local Pareto fronts over molecular nodes and applying it to objectives such as stock availability, step count, synthetic complexity, and route similarity. In contrast to our work, their approach relies on stochastic sampling and does not provide guarantees of Pareto-optimality or systematic front coverage. The original implementation of MO-MCTS concerns terminal objectives and thus renders a direct comparison to our algorithm difficult. Nevertheless, we introduce a variant of MO-MCTS for additive objectives and compare it to MORetro$^*$ in Appendix E.3. We refrain from drawing definitive conclusions from this comparison as no optimization of the MO-MCTS variant (*e.g.*, hyperparameters) was performed.

**Multi-objective A$^*$ Search** The problem of finding Pareto-optimal paths in search graphs has been studied extensively, with foundational algorithms including label-setting approaches for bi-criteria shortest paths (Martins, 1984; Stewart & White, 1991), and NAMOA$^*$ (Mandow & De La Cruz, 2010), extending to arbitrary numbers of objectives via dominance pruning of open labels. However, these methods operate on standard OR graphs, where node costs are path-independent. In retrosynthesis, the AND-OR structure introduces joint dependencies: reaction nodes require all reactant children to be simultaneously solved, meaning the cost of a molecule cannot be evaluated in isolation. We introduce a pruning mechanism for AND-OR graphs in Section 3.5, which generalizes NAMOA$^*$-style admissibility.

**Relevant Objectives for Retrosynthesis** Beyond simple objectives such as route length or number of building blocks, more informative criteria require additional information. Predicted reactions can be augmented with condition prediction models to estimate solvents, catalysts, and temperatures (Gao et al., 2018; Sun et al., 2025). This enables one to define objectives related to greenness, toxicity, and sustainability (Pu et al., 2019; Wang et al., 2020; Weber et al., 2021). Objectives reflecting scale-up potential and economic feasibility can further be defined using building-block prices (Hastedt et al., 2025) and reaction separability metrics (Kuznetsov & Sahinidis, 2021).

## 2.2. Problem Setting

**Synthesis Planning** We denote the set of all molecules as $\mathcal{M}$, the set of all reactions as $\mathcal{R}$, and the set of all transformation rules as $\mathcal{T}$. In this work, a transformation rule $\gamma_i \in \mathcal{T}$ is either a reaction template or a sequence of graph-edit instructions that can be applied to a product molecule $p_i \in \mathcal{M}$ to derive its corresponding set of reactants $r_i \subset \mathcal{M}$, such that

$$\gamma_i : \mathcal{M} \to 2^{\mathcal{M}}, \quad \gamma_i(p_i) = r_i. \tag{1}$$

We use $R_i \in \mathcal{R}$ to denote both a reaction node in the search graph and its associated reaction tuple $(r_i, a_i, T_i, p_i, \gamma_i)$, where $a_i \subset \mathcal{M}$ denotes the set of agents (reagents) and $T_i$ the reaction temperature. We assume the existence of a single-step retrosynthesis predictor $B$ and a reaction condition predictor $Q$. Given a product molecule $p_i$, the single-step predictor $B$ returns candidate reactant sets $r_i$ and transformation rules $\gamma_i$. The condition predictor $Q$ then augments each prediction with agents $a_i$ and temperature $T_i$, yielding the full reaction tuple:

$$R_i = (Q \circ B)(p_i) = (r_i, a_i, T_i, p_i, \gamma_i). \tag{2}$$

The goal of synthesis planning is to identify a synthesis route $\Gamma = \{R_1, \ldots, R_n\}$ such that all terminal reactants in the route, denoted by the frontier set $\mathcal{F}(\Gamma)$, are commercially available building blocks $\mathcal{BB}$, i.e., $\mathcal{F}(\Gamma) \subseteq \mathcal{BB} \subseteq \mathcal{M}$.

**Pareto Optimality** Let each reaction $R_i$ be associated with an $N_D$-dimensional non-negative cost $\mathbf{c}(R_i) \in \mathbb{R}_{>0}^{N_D}$, and let the cost of a synthesis route $\Gamma = \{R_1, \ldots, R_n\}$ be defined as $\mathbf{C}(\Gamma) = \sum_{i=1}^{n} \mathbf{c}(R_i)$. A synthesis route $\Gamma^*$ is Pareto optimal if there exists no other route $\Gamma$ such that $\mathbf{C}(\Gamma) \preceq \mathbf{C}(\Gamma^*)$ (component-wise) and $\mathbf{C}(\Gamma) \neq \mathbf{C}(\Gamma^*)$. The Pareto front $\mathcal{P}$ is defined as the set of all Pareto-optimal routes,

$$\mathcal{P} = \{\Gamma^* \mid \nexists \Gamma : \mathbf{C}(\Gamma) \preceq \mathbf{C}(\Gamma^*) \wedge \mathbf{C}(\Gamma) \neq \mathbf{C}(\Gamma^*)\}. \tag{3}$$

Let $\hat{\mathcal{P}}$ denote the set of non-dominated routes discovered by the algorithm during execution.

## 3. Methods

Our algorithm is built on previous work by Chen et al. (2020) (Retro$^*$) and extended by Xie et al. (2022) (RetroGraph).

## 3.1. A$^*$ Search for Multi-Objective Synthesis Planning

Following the A$^*$ formulation and Chen et al. (2020), the value $\mathbf{V}_t(m|G)$ of a molecule $m$ can be decomposed as

$$\mathbf{V}_t(m|G) = \mathbf{g}(m|G) + \mathbf{h}(m|G), \tag{4}$$

where $\mathbf{g}(m|G)$ and $\mathbf{h}(m|G)$ denote the $N_D$-dimensional cost of the path from the product root $t$ to $m$, and the estimated future synthesis cost, respectively. Thus, $\mathbf{V}_t(m|G)$

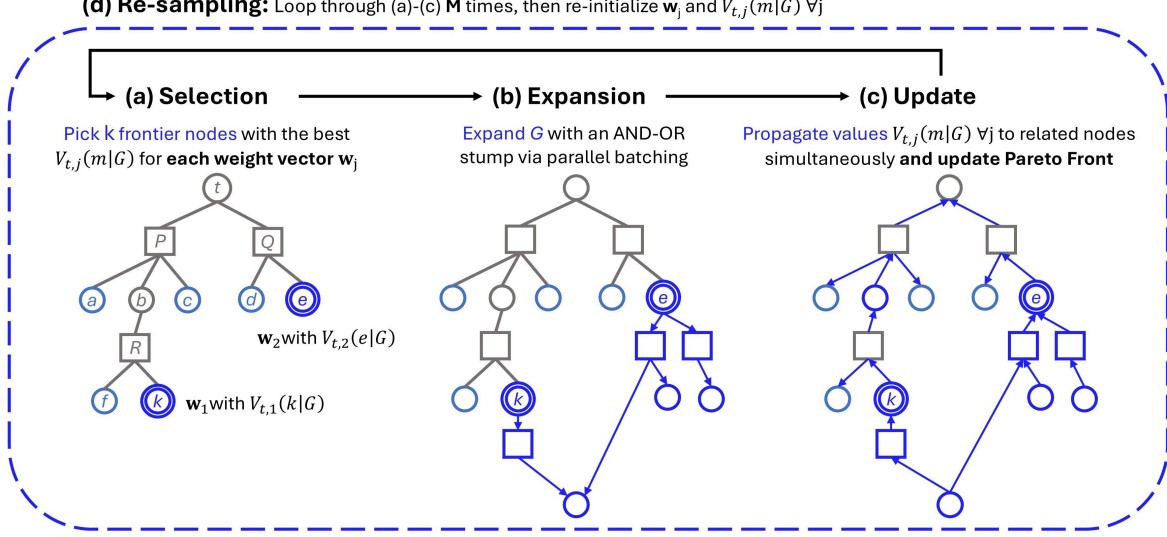

**(d) Re-sampling:** Loop through (a)-(c) **M** times, then re-initialize $\mathbf{w}_j$ and $V_{t,j}(m|G)$ ∀j

*Figure 2.* Overview of MORetro*. During each iteration, multiple frontier nodes are picked according to different weight vectors, expanded and new solutions are recorded. After $w_{\text{budget}}$ (M) iterations, weight vectors are re-sampled.

estimates the minimum cost of synthesizing the target product $t$ along routes that pass through $m$. We further introduce the reaction number $\mathbf{rn}(m|G)$ and the graph-independent heuristic estimate $\mathbf{V}_m$, both in $\mathbb{R}^{N_D}$. The heuristic $\mathbf{V}_m$ estimates the minimum cost required to synthesize molecule $m$ independently of the search graph $G$, where $\mathbf{V}_m = [V_{m,1}, \ldots, V_{m,N_D}]$ denotes one heuristic per objective. In contrast, the reaction number $\mathbf{rn}(m|G)$ estimates the minimum synthesis cost of $m$ conditioned on reactions currently present in $G$ (Chen et al., 2020) (see Eqs. 9–11).

To estimate reaction costs, we define a collection of scalar-valued cost functions $\{f_1, \ldots, f_{N_D}\}$, where each $f_i$ maps a reaction tuple $R$ to a scalar and corresponds to an individual objective. For a reaction $R_i$, these functions jointly define a vector-valued cost

$$\mathbf{c}(R_i) = (f_1(R_i), \ldots, f_{N_D}(R_i)) \in \mathbb{R}^{N_D}. \tag{5}$$

The exact objectives used in this study are further discussed in Section 3.4. Note that all costs are normalized to [0,1] using expected maximum and minimum values.

### 3.2. Overview of MORetro*

Our algorithm is presented visually in Figure 2 and in Algorithm 1. The algorithm is based on an AND-OR graph structure as proposed by Xie et al. (2022). The graph is a directed acyclic graph (DAG) to avoid any loops. Molecules are naturally represented as OR nodes (circles in Fig. 2), since only one child reaction must be solved, whereas reactions are represented as AND nodes (rectangles in Fig. 2) as all children molecules/reactants must be solved. Our algorithm is based on principled weight sampling, which

scalarizes the $N_D$-dimensional objective space into a single scalar objective. Specifically, we generate weight vectors $\mathbf{w}_j \in \mathbf{W}$ using a sampling mechanism $P(\mathbf{W})$, defined over the $N_D$-dimensional probability simplex,

$$\mathbf{W} = \left\{ \mathbf{w} \in \mathbb{R}^{N_D}_{\geq 0} \,\middle|\, \sum_{i=1}^{N_D} w_i = 1 \right\}. \tag{6}$$

The exact weight sampling strategies are further discussed in Section 3.3. We initially sample $N_S$ weight vectors $\{\mathbf{w}_j\}_{j=1}^{N_S}$ from $P(\mathbf{W})$. These weight vectors are used simultaneously during the search. In particular, for each sampled weight vector $\mathbf{w}_j \in \mathbf{W}$, we calculate the value function and reaction number via linear scalarization:

$$V_{t,j}(\cdot|G) = \mathbf{w}_j^\top \mathbf{V}_t(\cdot|G), \tag{7a}$$

$$rn_j(\cdot|G) = \mathbf{w}_j^\top \mathbf{rn}(\cdot|G), \quad \forall j \in \{1, \ldots, N_S\}. \tag{7b}$$

In other words, each weight vector induces its own scalarized value and reaction functions and conducts an independent search over the shared tree $G$. Once the weight vectors are sampled, MORetro* proceeds through four stages, outlined below. These stages build on ideas from Chen et al. (2020); Xie et al. (2022), with additional modifications to support multi-objective search. We therefore keep the description concise and refer the reader to prior work for detailed discussions.

**Selection** Given the frontier set $\mathcal{F}(G)$, each weight vector selects the molecule minimizing its scalarized value function,

$$m_j^{\text{next}} = \underset{m \in \mathcal{F}(G)}{\arg\min} V_{t,j}(m|G), \quad \forall j. \tag{8}$$

If multiple weights select the same molecule, they are grouped and processed jointly in subsequent steps.

**Algorithm 1** MORetro*$(t)$. All operations over $j$ are executed in parallel via batching; loop notation used for clarity.

---

**Input:** Graph $G = (\mathcal{V}, \mathcal{E})$ with $\mathcal{V} \leftarrow \{t\}$, $\mathcal{E} \leftarrow \varnothing$; $\hat{\mathcal{P}} \leftarrow \varnothing$; Weights $\mathbf{W}$; Budget $w_{\text{budget}}$
**Initialize:** Sample weights $\{\mathbf{w}_j\}_{j=1}^{N_S} \sim P(\mathbf{W})$; $k \leftarrow 0$
**repeat**
    **for all** $j = 1, \dots, N_S$ **do**
        $m_j^{\text{next}} \leftarrow \arg\min_{m \in \mathcal{F}(G)} V_{t,j}(m|G)$ {Selection}
        $\{R_{i,j}\}_{i=1}^{K} \leftarrow (Q \circ B)(m_j^{\text{next}})$ {Prediction}
        $\{\mathbf{c}(R_{i,j})\}_{i=1}^{K} \leftarrow (f_1(R_{i,j}), \dots, f_{N_D}(R_{i,j}))$
        **for all** $i = 1, \dots, K$ **do**
            {Add reaction and reactant nodes}
            $G \leftarrow G \cup \{R_{i,j}, r_{i,j}\}$
        **end for**
    **end for**
    **Update** value functions $\{V_{t,j}\}_{j=1}^{N_S}$ on $\mathcal{F}(G)$
    **Record** any new non-dominated route by updating $\hat{\mathcal{P}} \leftarrow \text{ND}(\hat{\mathcal{P}} \cup \Gamma^*)$ and the successful weight $\mathbf{w}_j$
    **if** $k > 0$ **and** $k \bmod w_{\text{budget}} = 0$ **then**
        **Reinitialize** weights $\{\mathbf{w}_j\}_{j=1}^{N_S} \sim P(\mathbf{W})$
        **Update** value functions $\{V_{t,j}\}_{j=1}^{N_S}$ on $\mathcal{F}(G)$
    **end if**
    $k \leftarrow k + 1$
**until** termination = True
**return** Pareto front $\hat{\mathcal{P}}$

---

**Expansion** The single-step retrosynthesis and condition predictors $B(\cdot)$ and $Q(\cdot)$ are applied sequentially to the selected molecules $m_j^{\text{next}}$ using batched inference. This renders our implementation computationally efficient. For each $m_j^{\text{next}}$, the models predict $K$ candidate reactions $\{R_{i,j}\}_{i=1}^{K}$ with associated cost vectors $\{\mathbf{c}(R_{i,j})\}_{i=1}^{K}$. All resulting reaction and molecule nodes are added to the search tree $G$, and newly introduced molecule nodes are initialized as

$$rn_j(m|G) \leftarrow \mathbf{w}_j^\top \mathbf{V}_m, \quad \forall j. \qquad (9)$$

**Update** Following Yu et al. (2024), costs are propagated upward through the graph for all weights $\mathbf{w}_j$. For a reaction node $R$, the reaction number is computed as

$$rn_j(R|G) \leftarrow \mathbf{w}_j^\top \mathbf{c}(R) + \sum_{m \in ch(R)} rn_j(m|G). \qquad (10)$$

For non-frontier molecule nodes,

$$rn_j(m|G) \leftarrow \min_{R \in ch(m)} rn_j(R|G). \qquad (11)$$

These updates are applied recursively until the target node $t$ is reached, with $V_{t,j}(t|G) \leftarrow rn_j(t|G)$. Value updates are

then propagated downward as

$$V_{t,j}(R|G) \leftarrow rn_j(R|G) - rn_j(pr(R)|G) + V_{t,j}(pr(R)|G),$$
$$V_{t,j}(m|G) \leftarrow \min_{R \in pr(m)} V_{t,j}(R|G), \qquad (12)$$

where $ch(\cdot)$ and $pr(\cdot)$ denote children and parent nodes, respectively. Finally, each newly found Pareto-optimal route $\Gamma^*$ is stored with its corresponding hypervolume contribution and the weight vector that generated it.

**Re-sampling** Every $w_{\text{budget}}$ iterations, $N_S$ new weights are drawn from $P(\mathbf{W})$. Following re-sampling, the reaction numbers $rn_j(\cdot|G)$ and value functions $V_{t,j}(\cdot|G)$ are updated for all nodes in the graph. The algorithm runs until termination[1]. We next describe the sampling strategies investigated.

### 3.3. Weight Sampling Strategies

We first investigate two simple weight-sampling strategies: grid sampling and Sobol sampling. For deterministic grid sampling, weights are drawn from a uniform grid:

$$\mathbf{w} \sim P(\mathbf{W}) = \text{Uniform}(\mathbf{W}_{\text{grid}}), \qquad (13)$$

where $\mathbf{W}_{\text{grid}} \subseteq \mathbf{W}$ is a finite set of weight vectors forming a uniform grid. For Sobol sampling, we generate $N_S$ quasi-random low-discrepancy weight vectors that evenly cover the space:

$$\mathbf{w} \sim P(\mathbf{W}) = \text{Sobol}(\mathbf{W}). \qquad (14)$$

On top of these simple strategies, we propose a solution-informed Bayesian optimization (BO) weight-sampling scheme (Figure 3). During a warm-up phase, weights are

---

[1]Termination is triggered if (i) time budget or (ii) single-step budget is exhausted, or if (iii) all weights have been sampled.

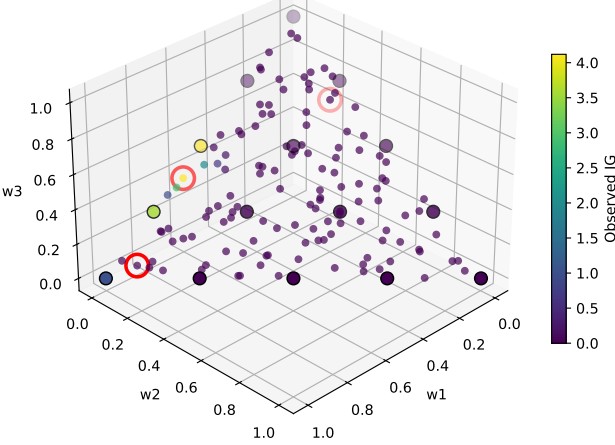

*Figure 3.* BO-informed sampling. The selected weights (drawn from Sobol sequences) for the next iteration are encircled. Larger dots show weights that were explored during the warm-up period.

drawn from the grid or Sobol sequence (discussed further in Appendix D). After warm-up, $P(\mathbf{W})$ is defined implicitly via BO: a surrogate model is fit to the observed hypervolume (HV) improvements of previously evaluated weights, and new weights are sampled to maximize expected information gain (IG). Since MORetro$^*$ evaluates weights in batches of size $N_S$, we use the GIBBON acquisition function $\alpha_{\text{GIBBON}}$ (Moss et al., 2021), which approximates the information gain of jointly evaluating a batch while promoting diversity. The resulting sampling mechanism satisfies

$$\mathbf{w} \sim P(\mathbf{W}) \propto \exp\!\Big(\alpha_{\text{GIBBON}}\big(\{\mathbf{w}_j\}_{j=1}^{N_S}\big)\Big). \quad (15)$$

### 3.4. Proposed Objectives for Synthesis Planning

While any set of objectives could be used for multi-objective planning, we propose three that capture practical considerations in chemical synthesis: **sustainability**, **toxicity (hazardousness)**, and **scale-up potential**. Sustainability is quantified as a linear combination of reaction temperature and atom economy. Toxicity accounts for the hazards associated with catalysts and solvents. Scale-up potential measures the "ease of separability" of the reaction mixture, approximated as the logP difference between products and reactants. Together, these objectives contribute to $\mathbf{g}(m|G)$, representing the cumulative cost of a synthesis pathway $\Gamma$. To estimate the future heuristic cost $\mathbf{V}_m$, we avoid training offline models on a limited set of existing pathways, as done previously (Chen et al., 2020), since such models exhibit high variance and provide minimal performance gain. Instead, we define heuristics aligned with the proposed objectives. For instance, for toxicity, we predict the toxicity of molecule $m$ (Pu et al., 2019): if $m$ contains toxic moieties, its expansion is discouraged, as future reactants are likely to inherit these.

Finally, we introduce an additional guiding objective designed solely to steer the search toward purchasable building blocks. This objective is *not included* in the Pareto analysis (metrics reported in Section 4 are computed over the remaining three objectives). Examples include reaction probability and the value estimate heuristic from Retro$^*$. A complete description of all objectives is provided in Appendix C.

### 3.5. Optimality Guarantee for the Pareto Front

In this section, we introduce a tight, vector-valued quantity on the cost of synthesizing the target molecule $t$ through an intermediate molecule $m$. This quantity:

1. Guarantees correctness of the recovered Pareto front under our search procedure.

2. Enables safe pruning of unpromising frontier molecules $m \in \mathcal{F}(G)$.

The full proof is provided in Appendix B.

**Definition 3.1** (Pruning Number). The vector-valued *pruning number* $\mathbf{pn}$ for each node in a retrosynthesis AND-OR graph is defined recursively as:

$$\mathbf{pn}(R|G) = \mathbf{c}(R) + \sum_{m \in ch(R)} \mathbf{pn}(m|G),$$
$$\mathbf{pn}(m|G) = \min_{R \in ch(m)} \mathbf{pn}(R|G), \quad \forall m \notin \mathcal{F}(G), \quad (16)$$
$$\mathbf{pn}(m|G) = \mathbf{V}_m, \quad \forall m \in \mathcal{F}(G),$$

where all sums and minima are taken component-wise. The component-wise minimum preserves the lower bound.

**Definition 3.2** ($\mathbf{V}_{\text{bound}}$). The vector-valued $\mathbf{V}_{\text{bound}}$ is computed from $\mathbf{pn}$ using the same propagation rules as in Eq. 12 (substitute $\mathbf{pn}$ for $rn_j$ and $\mathbf{V}_{\text{bound}}$ for $V_{t,j}(m)$). It estimates the cost of any *hypothetical* synthesis route passing through a molecule $m$.

**Definition 3.3** (Bound dominance condition). For an unexpanded molecule $m \in \mathcal{F}(G)$, we say that the *bound dominance condition* holds if there exists a discovered synthesis route $\hat{\Gamma} \in \hat{\mathcal{P}}$ such that

$$\mathbf{C}(\hat{\Gamma}) \prec \mathbf{V}_{\text{bound}}(m). \quad (17)$$

**Theorem 3.4.** *Let $G$ be a retrosynthesis AND-OR graph with reaction costs $\mathbf{c}(R) \in \mathbb{R}_{\geq 0}^{N_D}$. Assume that $\mathbf{V}_m$ or a valid lower bound is known for all molecules $m$, and that at least one feasible synthesis route exists (i.e., $\mathcal{P} \neq \varnothing$).*

*Then, if the termination criterion is modified to halt when the bound dominance condition (Definition 3.3) holds for all $m \in \mathcal{F}(G)$, Algorithm 1 returns $\hat{\mathcal{P}} = \mathcal{P}$.*

**Corollary 3.5** (Safe node pruning). *Any unexpanded molecule $m \in \mathcal{F}(G)$ satisfying the bound dominance condition cannot lie on a Pareto-optimal synthesis route and may therefore be safely pruned.*

*Remark* 3.6. If the dominance relation in the bound dominance condition is relaxed to $\varepsilon$-dominance, the theorem extends directly to guarantee recovery of the $\varepsilon$-Pareto front.

Since the heuristics defined in Section 3.4 are not admissible, we use the zero vector $\mathbf{V}_m = [0, \ldots, 0] \in \mathbb{R}^{N_D}$ to calculate $\mathbf{V}_{\text{bound}}$, in practice.

## 4. Experiments

Our experiments are designed to answer the following: **(1)** Does MORetro$^*$ improve the performance of multi-objective synthesis planning compared to baseline methods in terms of Pareto front quality? **(2)** Does MORetro$^*$ maintain the success rate (finding at least one solution) compared to the single-objective baseline? **(3)** By embedding multiple conflicting objectives, does MORetro$^*$ improve diversity for Pareto-optimal synthesis routes in terms of reaction chemistry? **(4)** Which weight sampling strategy

*Table 1.* Summary of multi-objective performance on the USPTO-190 dataset for three different single-step models. Our algorithm with BO sampling is compared to the Fixed and single-objective Retro* baselines. The number of single-step expansions is kept to $N_B = 300$.

| Model | Method | Pareto front Statistics | | MORetro* Dominance | |
|---|---|---|---|---|---|
| | | HV ($\uparrow$) | R2 ($\downarrow$) | Base. Dom. % ($\uparrow$) | Self Dom. % ($\downarrow$) |
| G2E | Retro* | $0.98 \pm 0.32$ | $0.45 \pm 0.14$ | 34% | 6% |
| | Fixed | $0.97 \pm 0.40$ | $0.47 \pm 0.09$ | 21% | 2% |
| | MORetro* (BO) | $\mathbf{1.04 \pm 0.31}^{***}$ | $\mathbf{0.43 \pm 0.11}^{***}$ | - | - |
| PDVN | Retro* | $0.88 \pm 0.37$ | $0.48 \pm 0.12$ | 26% | 8% |
| | Fixed | $0.74 \pm 0.54$ | $0.5 \pm 0.12$ | 36% | 2% |
| | MORetro* (BO) | $\mathbf{0.92 \pm 0.37}^{***}$ | $\mathbf{0.47 \pm 0.11}^{*}$ | - | - |
| NeuralSym | Retro* | $0.52 \pm 0.51$ | $0.49 \pm 0.13$ | 31% | 17% |
| | Fixed | $0.48 \pm 0.55$ | $0.51 \pm 0.09$ | 18% | 2% |
| | MORetro* (BO) | $\mathbf{0.56 \pm 0.51}^{***}$ | $\mathbf{0.48 \pm 0.10}$ | - | - |

Statistical significance MORetro* vs. baselines: *** $p < 8 \times 10^{-3}$ (Bonferroni), ** $p < 0.01$, *$p < 0.05$; Wilcoxon Test

(if any) is preferred to guide the search? **(5)** What is the practical gain of search space pruning and how effective is the optimality guarantee?

### 4.1. Experimental Setup

**Datasets** We select different datasets to show that our method is not biased towards a specific data source. First, we benchmark our methodology on the widely-used USPTO-190 dataset (Chen et al., 2020), a set of 190 molecular targets from the USPTO-full dataset. We also use the Pistachio-reachable dataset from Yu et al. (2024) and a random selection of 150 drug-like molecules from ChEMBL (Wang & Montana, 2025) (results for these datasets are shown in Appendix E).

**Single-Step Algorithms** We select three single-step models to demonstrate that our method is independent of the underlying predictor: the template-based NeuralSym (Yu et al., 2024) and PDVN models (Liu et al., 2023; Hassen et al., 2026), and the template-free Graph2Edits model from InterRetro (Wang & Montana, 2025). The models predict $K = 25$ (or $K = 50$ for PDVN) reactions per molecule.

**Multi-step Baselines** In the main body, we compare MORetro* to single-objective Retro* (Chen et al., 2020) and a scalarized baseline using a fixed weight vector ($\mathbf{w}_{\text{fixed}} = [.2, .2, .2, .4]$), placing more emphasis on the guidance objective. We refer to this baseline as *Fixed*. To show that *Fixed* is not cherry-picked, we present results for additional weight combinations in Appendix E.2. For Retro*, we construct the Pareto front by extracting all synthesis routes $\{\Gamma_1, \ldots, \Gamma_l\}$ at search termination, computing their costs $\mathbf{C}(\Gamma)$, and extracting the front in $N_D$ dimensions. We impose a budget of 300 single-step expansions for all methods. Building blocks come from 23M commercially available molecules from *eMolecules* (Chen et al., 2020). Hyperparameter values for all weight-sampling strategies are provided in Appendix D.

The comparison to the MO-MCTS variant is shown in Appendix E.3.

**Computational Resources** All experiments were conducted on a workstation equipped with an NVIDIA RTX6000 and an AMD EPYC 7742. On average, the search terminates in 6–12 minutes depending on the single-step model.

### 4.2. Results

We evaluate solution quality primarily using hypervolume (HV), which measures dominated objective space, and the R2 indicator (Brockhoff et al., 2012) to capture expected trade-off quality. Higher HV indicates better convergence toward the ideal point (zero cost), while lower R2 reflects improved Pareto coverage. Molecules without a synthesis route are assigned HV = 0 and excluded from R2 analysis. All objectives are normalized using the 5th and 95th percentiles of the costs across all obtained synthesis routes. HV and R2 are computed with respect to the reference point $[1.1, 1.1, 1.1]$. We also report dominance coverage: *Base-*

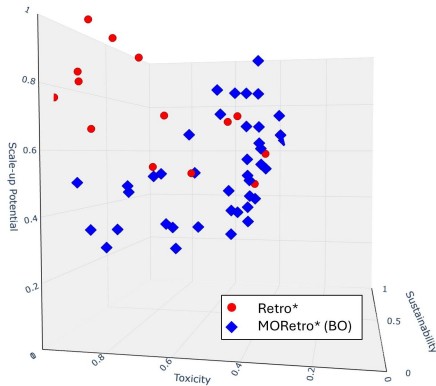

*Figure 4.* Visual comparison of two Pareto fronts returned by Retro* and MORetro* with hypervolume difference of $\Delta$HV=0.07

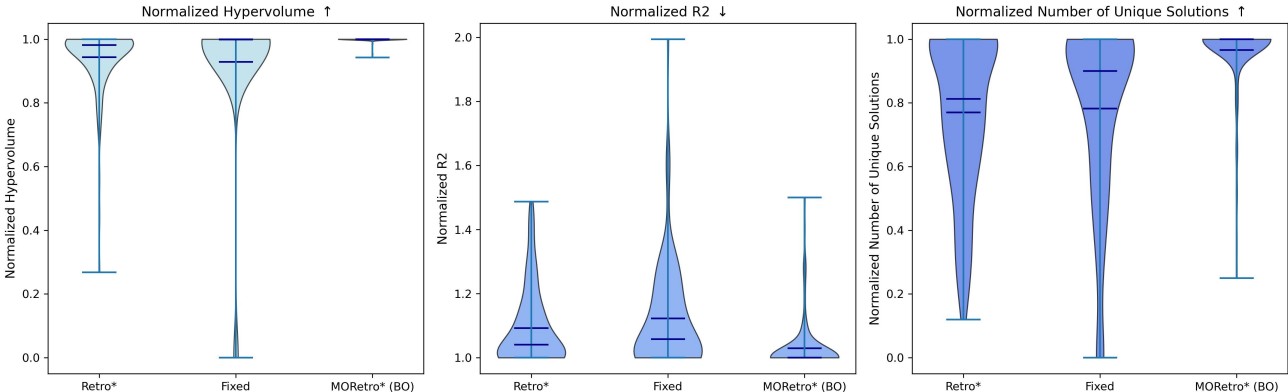

*Figure 5.* Per-molecule comparison of Pareto front metrics for the ChEMBL (G2E) experiment. Statistics are normalized according to the best value found per molecule.

*line Dominance* (%) is the fraction of baseline solutions dominated by MORetro*, and *Self Dominance* (%) is the fraction of MORetro* solutions dominated by the baseline.

To address **(1)**, Table 1 summarizes the Pareto statistics. MORetro* consistently outperforms both Retro* and the Fixed baseline in terms of HV and R2, achieving HV improvements of approximately 5–8%. All methods exhibit substantial variance, which is expected given the discrete nature of synthesis planning and the wide range of achievable objective values across molecules. To assess statistical significance, we perform paired Wilcoxon signed-rank tests on a per-molecule basis, comparing MORetro* against each baseline. All HV improvements are statistically significant with p-values $< 0.01$. Figure 5 visually corroborates these findings, showing normalized per-molecule performance relative to the best observed value. Notably, MORetro* also produces the largest number of unique Pareto-optimal routes on average. In terms of dominance, MORetro* dominates approximately 18–36% of baseline solutions, while only 2–8% of its solutions are dominated by the baselines. Finally, Figure 4 shows a HV improvement of roughly 10%, visually confirming the significance of such an improvement. Results on additional datasets (Pistachio-reachable and ChEMBL) reported in Appendix E.1 confirm these trends.

**Success Rate** Table 2 addresses **(2)**. MORetro* exhibits only a marginal reduction in success rate compared to Retro*, failing on just 2–3 additional molecules out of 190. In contrast, the Fixed baseline solves up to 25% fewer targets. This highlights the importance of dynamically balancing the guidance objective: while the Fixed baseline assigns a constant weight of 0.4, MORetro* adaptively samples weights that emphasize guidance when needed, resulting in substantially higher success rates.

**Diversity of Solutions** To answer **(3)**, we evaluate the chemical diversity of Pareto-optimal synthesis routes using a recently proposed pairwise similarity metric (Genheden & Shields, 2025). We report both the average maximum dissimilarity (DS) and the diversity fraction, defined as the proportion of route pairs with dissimilarity exceeding 0.5. Statistical significance is assessed using the same Wilcoxon test as above. As shown in Table 3, MORetro* consistently achieves higher diversity than both baselines, often with strong statistical significance. These results demonstrate that explicitly optimizing multiple objectives naturally leads to better exploration and more diverse synthesis plans.

*Table 2.* Success rate (finding at least 1 synthesis route per molecule) of MORetro* versus baselines on USPTO-190

| Model | Method | Success Rate (%) | ΔRetro* % |
|---|---|---|---|
| G2E | Retro* | 100 | - |
| | Fixed | 90.5 | -9.5 |
| | MORetro* (BO) | 97.9 | -2.1 |
| PDVN | Retro* | 98.9 | - |
| | Fixed | 69.8 | -29.1 |
| | MORetro* (BO) | 96.8 | -2.1 |
| NeuralSym | Retro* | 61.6 | - |
| | Fixed | 47.4 | -14.2 |
| | MORetro* (BO) | 60.0 | -1.6 |

*Table 3.* Chemical diversity (div.) of Pareto-optimal synthesis routes for USPTO-190. Max. DS is the maximum dissimilarity between a pair of routes.

| Model | Method | Max. DS (↑) | Div. Frac. (↑) |
|---|---|---|---|
| G2E | Retro* | 0.80±0.26 | 0.43±0.26 |
| | Fixed | 0.77±0.29 | 0.41±0.27 |
| | MORetro* (BO) | **0.84**±0.23** | **0.49**±0.23** |
| PDVN | Retro* | 0.72±0.29 | 0.38±0.29 |
| | Fixed | 0.67±0.31 | 0.34±0.30 |
| | MORetro* (BO) | **0.76**±0.27** | **0.41**±0.28 |
| NeuralSym | Retro* | 0.57±0.31 | 0.24±0.28 |
| | Fixed | 0.58±0.33 | 0.28±0.30 |
| | MORetro* (BO) | **0.64**±0.31* | **0.32**±0.29** |
| Statistical sign.: ** $p < 0.01$, * $p < 0.05$; Wilcoxon Test | | | |

*Table 4.* Search space reduction and PF quality when using $\mathbf{V}_{\text{bound}}$ for pruning on USPTO-190 (G2E). "PO – 100% cut": number (%) of molecules whose frontier is fully pruned, certifying Pareto optimality. 95% and 90% columns analogous. $\varepsilon$-$\mathbf{V}_{\text{bound}}$ uses $\varepsilon = 0.1$. Deltas for PF Quality (HV and R2) are relative to MORetro* (BO) in Table 1, with asterisks denoting the same statistical significance levels.

| Methods | Search Space Reduction in # mols | | | Search Space | PF Quality | | Runtime |
| | PO – 100% cut | 95% cut | 90% cut | Average Reduction (%) | HV (↑) | R2 (↓) | Model calls |
|---|---|---|---|---|---|---|---|
| $\mathbf{V}_{\text{bound}}$ | 11 (6%) | 17 (9%) | 39 (21%) | $54 \pm 33$ | 1.045* (+0.005) | 0.41*** (-0.03) | $289 \pm 50$ |
| $\varepsilon$-$\mathbf{V}_{\text{bound}}$ | 32 (17%) | 40 (21%) | 65 (34%) | $61 \pm 35$ | 1.045* (+0.005) | 0.43* (-0.01) | $265 \pm 83$ |

**Sampling Strategy** We investigate **(4)** through an ablation study comparing the BO-based sampling strategy to Sobol and grid-based sampling (Tables 9 and 10, Appendix E). BO sampling consistently outperforms Sobol sampling across PF quality, success rate, and diversity. Comparisons with grid sampling are more nuanced: BO achieves clear gains in HV and success rate in two experiments, while performance is comparable in the remaining cases. This suggests that grid sampling can serve as a strong baseline, although BO sampling provides more robust improvements overall.

**Pruning and Optimality Guarantee** (**5**) Finally, we show the practical gain of using $\mathbf{V}_{\text{bound}}$ in Table 4. The preliminary results for the USPTO-190 dataset show that pruning can lead to an effective reduction of the search space. Using the exact $\mathbf{V}_{\text{bound}}$ to prune open nodes, an average search space reduction of 54% is observed. Notably, 21% of the molecules have their search space reduced by 90% (*i.e.*, 90% of open nodes are pruned) upon termination. We observe an improvement in the PF quality, with a statistically lower R2 score. Table 4 demonstrates that one can obtain a more aggressive pruning strategy by using an approximate $\varepsilon$-$\mathbf{V}_{\text{bound}}$. By pruning open nodes whose $\mathbf{V}_{\text{bound}}$ is dominated up to a slack of $\varepsilon$, the search space is on average decreased by 60% upon termination. Furthermore, for 32 molecules, the algorithm certifies the $\varepsilon$-PF without deteriorating the PF quality compared to Table 1. This also translates to a lower overall runtime with 265 single-step model calls on average.

## 5. Limitations and Future Outlook

MORetro* assumes that reaction costs are additive, currently limiting objectives to those that decompose additively per reaction or molecule. A principled extension could use admissible additive lower bounds to guide search while evaluating the true non-additive cost only upon route completion, preserving theoretical guarantees via pruning. Furthermore, the practical heuristics currently employed are not provably admissible, meaning the pruning bound falls back to the zero vector in practice, reducing pruning efficiency. Constructing tight admissible lower bounds for such objectives remains an open challenge. In addition, the current study fixes the number of Pareto objectives at three; investigating how the algorithm scales to a larger set of diverse objectives is a natural next step. Future work will also address the ro-

bustness and chemical accuracy of the proposed objectives to support broader adoption of CASP within industry, as well as developing adaptive strategies to close the remaining success rate gap to single-objective search.

## 6. Conclusion

In this work, we introduce MORetro*, a novel framework for multi-objective retrosynthetic planning. By combining weight scalarization with Bayesian optimization-informed sampling, MORetro* efficiently recovers high-quality Pareto fronts of synthesis routes under a fixed expansion budget. Across multiple datasets and single-step prediction models, our approach consistently improves Pareto front quality and chemical diversity over single-objective and fixed weight baselines, while largely preserving their success rates. By directly incorporating practical objectives into the planning process, MORetro* provides a flexible and effective foundation for accelerating decision-making, discovery, and the scale-up of industrial chemical synthesis.

## Code Availability

The code repository can be accessed via GitHub: https://github.com/OptiMaL-PSE-Lab/MORetro.

## Acknowledgements

We would like to thank the reviewers for the insightful discussions during peer review. We thank Emma Pajak and Mathias Neufang for proof-reading. Friedrich Hastedt thanks Alex Ganose for helpful discussions. Friedrich Hastedt acknowledges support from the Engineering and Physical Sciences Research Council (EPSRC) funding grant EP/S023232/1. We acknowledge computational resources and support provided by the Imperial College Research Computing Service (http://doi.org/10.14469/hpc/2232).

## Impact Statement

This paper presents work whose goal is to advance the field of machine learning. There are many potential societal consequences of our work, none of which we feel must be specifically highlighted here.

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

## A. Single-Step Retrosynthesis and Condition Prediction Models

This section provides a short conceptual description of the models utilized in this study.

**NeuralSym / PDVN** treat one-step retrosynthesis as a template retrieval task. Given a product molecule and a library of templates, the model predicts a categorical distribution over all templates in the dataset, then applies up to $K$ templates to the product. Following the reaction rule, the product is converted into a set of reactants. NeuralSym serves as the base template model. PDVN adds a second neural network trained via reinforcement learning to reweight the likelihood of templates proposed by the base model (Liu et al., 2023). In this paper, we refer to this reweighted likelihood of reaction templates as the *policy cost*.

**Graph2Edits (G2E)** treats one-step retrosynthesis as a graph-editing sequence prediction task (Zhong et al., 2023). Starting from the product molecule, the model autoregressively predicts the next action to apply to the molecular graph. The pre-defined edit vocabulary includes: Delete Bond, Change Bond, Change Atom, Attach Leaving Group, and Terminate. The model uses a Graph Neural Network (GNN) to predict the next action and requires a pre-defined leaving group vocabulary. The InterRetro framework trains G2E via reinforcement learning to reweight the likelihood of reactants and improve multi-step search success (Wang & Montana, 2025). We refer to the predicted likelihood from this model as the *policy cost*.

**QUARC** is a reaction condition prediction model (Sun et al., 2025). Given a product and corresponding reactants, it predicts temperature, agent identity (catalysts and solvents), and agent loading. The model is based on a GNN. In this work, we disregard agent loading and only use the prediction for temperature and agent identity.

## B. Proof of Theorem 3.4

We prove that Algorithm 1, with the termination criterion based on the bound dominance condition (Definition 3.3), returns $\hat{\mathcal{P}} = \mathcal{P}$, *i.e.*, the true Pareto front.

### B.1. Connection to single-objective Retro*

Retro* (Chen et al., 2020) is a single-objective A* algorithm applied to retrosynthesis AND-OR graphs. In this setting, admissibility of the heuristic $V_m$ guarantees that Retro* returns an optimal solution, which also implies that $V_t(m)$ is itself a valid lower bound on the cost of any synthesis route passing through $m$.

**Lemma B.1** (Retro* admissibility, single-objective (Chen et al., 2020)). *Let $V_m$ be a valid lower bound for all molecules $m$. Then Retro* is guaranteed to return an optimal solution if it terminates when the cost of a found route does not exceed*

$$\min_{m \in \mathcal{F}(G)} V_t(m).$$

### B.2. Multi-objective admissibility of $\mathbf{V}_{\text{bound}}$

**Lemma B.2** (Admissibility of $\mathbf{V}_{\text{bound}}$). *Assume that for all frontier molecules $m \in \mathcal{F}(G)$, a valid lower bound $\mathbf{V}_m$ is known. Then, for any molecule $m$,*

$$\mathbf{V}_{bound}(m|G) \preceq \mathbf{C}(\Gamma),$$

*for any feasible synthesis route $\Gamma$ passing through $m$. In other words, $\mathbf{V}_{bound}$ is component-wise admissible.*

*Proof.* Our definitions of $\mathbf{V}_{\text{bound}}$ and $\mathbf{pn}$ generalize the single-objective $V_t$ and $rn$ in Retro* to $N_D$ objectives such that:

$$\mathbf{V}_{\text{bound}} = [V_{t,1}, \ldots, V_{t,N_D}].$$

By Lemma B.1, admissibility holds for each individual objective.

Moreover, one can easily show that $\mathbf{V}_{\text{bound}}$ is component-wise admissible: by induction from the root node $t$ to all descendant reactions and molecules:

*Root node:* $\mathbf{V}_{\text{bound}}(t|G) = \mathbf{pn}(t|G)$ is admissible (component-wise). From Equation 16, $\mathbf{pn}$ at each node is recursively computed as the sum over reaction costs and child pruning numbers. At frontier molecules $\mathbf{pn}(m|G) = \mathbf{V}_m$ is an admissible lower bound, by assumption. Thus, $\mathbf{pn}(\cdot)$ is admissible for all nodes in $G$.

*Reaction nodes:* For a reaction $R$,

$$\mathbf{V}_{\text{bound}}(R|G) = \mathbf{pn}(R|G) - \mathbf{pn}(pr(R)|G) + \mathbf{V}_{\text{bound}}(pr(R)|G),$$

the subtraction isolates the cost contribution of $R$ and its descendants. Since both $\mathbf{pn}$ and $\mathbf{V}_{\text{bound}}(pr(R)|G)$ are admissible, so is $\mathbf{V}_{\text{bound}}(R|G)$.

*Molecule nodes:* For a molecule $m$,

$$\mathbf{V}_{\text{bound}}(m|G) = \min_{R \in pr(m)} \mathbf{V}_{\text{bound}}(R|G),$$

the component-wise minimum preserves admissibility because any feasible route must pass through at least one parent reaction.

By induction, $\mathbf{V}_{\text{bound}}$ is component-wise admissible for all nodes. $\qquad\square$

### B.3. Proof of Theorem 3.4

*Proof sketch.* Let $\hat{\mathcal{P}}$ denote the set of Pareto-optimal routes discovered at termination.

**Soundness.** All routes in $\hat{\mathcal{P}}$ are explicitly constructed and non-dominated, so they are Pareto-optimal.

**Completeness.** Suppose a Pareto-optimal route $\Gamma^* \notin \hat{\mathcal{P}}$. Then $\Gamma^*$ must pass through at least one frontier molecule $m \in \mathcal{F}(G)$ at termination. By the termination criterion, there exists a discovered route $\hat{\Gamma} \in \hat{\mathcal{P}}$ such that

$$\mathbf{C}(\hat{\Gamma}) \prec \mathbf{V}_{\text{bound}}(m).$$

By Lemma B.2, $\mathbf{V}_{\text{bound}}(m)$ lower-bounds the cost of any route through $m$, including $\Gamma^*$:

$$\mathbf{C}(\hat{\Gamma}) \prec \mathbf{V}_{\text{bound}}(m) \preceq \mathbf{C}(\Gamma^*).$$

By transitivity, $\mathbf{C}(\hat{\Gamma}) \prec \mathbf{C}(\Gamma^*)$, meaning $\hat{\Gamma}$ dominates $\Gamma^*$. This contradicts the Pareto-optimality of $\Gamma^*$. Hence, all Pareto-optimal routes are discovered, proving completeness. $\qquad\square$

**Remark.** Since reaction costs are non-negative, the zero vector $\mathbf{0}$ is always a valid lower bound for $\mathbf{V}_m$ for any molecule $m$.

## C. Objective Functions

This section provides a detailed description of the objectives used throughout this work.

**Sustainability Objective.**
The sustainability objective is defined as the average of two components: reaction temperature and a simplified atom economy metric. Atom economy is computed as the ratio of heavy atoms in the product molecule to the total number of heavy atoms in the reactant molecules,

$$AE = \frac{p_{\text{atoms}}}{r_{\text{atoms}}}. \tag{18}$$

Reaction temperature is incorporated via a piecewise penalty function $C(T)$ that reflects increasing energetic cost outside ambient conditions:

$$C(T) = \begin{cases} 0, & 15 \leq T \leq 25, \\ 0.25, & 10 \leq T < 15 \text{ or } 25 < T \leq 40, \\ 0.6, & -20 \leq T < 10, \\ 1.0, & T < -20, \\ 0.4, & 40 < T \leq 120, \\ 0.8, & T > 120. \end{cases} \tag{19}$$

The overall sustainability cost of a reaction $R_i$ is then defined as

$$C_{\text{Sust}}(R_i) = 0.5\, C(T) + 0.5\, (1 - AE), \tag{20}$$

where lower values correspond to more sustainable reactions.

Estimating the future cost is extremely difficult. Consequently, we employ a simple synthetic accessibility metric (SAScore (Ertl & Schuffenhauer, 2009)). The metric acts as a proxy for the number of reactions remaining towards building blocks. The more reactions remaining, the more likely it is to incur higher cost of synthesis (also in terms of operational sustainability, each new reaction will require additional operating costs). The heuristic is thus calculated as:

$$V_{m,\text{Sust}} = \frac{\text{SAScore}(m)}{10},\tag{21}$$

The SAScore is normalized between 0–1 by dividing it by its upper bound (10).

**Toxicity (Hazardousness).**
The toxicity objective captures the hazardousness of a reaction through the toxicity of auxiliary agents, including catalysts and solvents. Here, toxicity refers to both adverse effects on human health and environmental harm. Each agent provided by the QUARC predictor is assigned a toxicity cost based on external references. In particular, solvent toxicity scores are derived from the solvent selection and recommendation guide by Prat et al. (2016), while catalyst toxicity scores are informed by a recent study on the greenness and toxicity of heterogeneous catalysts (Bystrzanowska et al., 2019). Using these sources, toxicity costs were assigned to $\sim$150 out of 1,375 agents.

To score the remaining agents, we leverage GPT-4o, following prior work demonstrating the effectiveness of large language models for chemical property annotation tasks (Mirza et al., 2025). A high-level description of the prompting procedure is provided in Appendix F. All agent toxicity costs are normalized to the interval $[0, 1]$, where non-hazardous agents receive a cost of 0 and the most hazardous agents receive a cost of 1. The resulting reaction-level toxicity cost is denoted by $C_{\text{Tox}}(R_i)$.

In addition to agent toxicity, we account for the intrinsic toxicity of reactant molecules. Since accurately estimating the toxicity of future intermediates is virtually impossible, we instead penalize the presence of toxic moieties in reactants that do not contribute to the final product. This reflects the intuition that toxic substructures are likely to persist across building blocks and precursors. We quantify molecular toxicity using the EToxPred model (Pu et al., 2019), defining the toxicity value of a molecule $m$ as

$$V_{m,\text{Tox}} = \text{EToxPred}(m),\tag{22}$$

which corresponds to the predicted probability of the molecule being toxic.

**Scale-up Potential.**
This objective captures the feasibility of scaling a reaction to industrial settings, where major cost drivers include downstream separation and the cost of starting materials. To approximate separation difficulty, we adopt a common heuristic based on the partition coefficient $\log(P)$, which is widely used to assess extractive separability. Specifically, we employ a learned $\log(P)$ predictor and compute the *ExtractionScore* of a reaction mixture following Kuznetsov & Sahinidis (2021).

For a given reaction, we compute the absolute $\log(P)$ differences between the product and each reactant (solvents are currently ignored but could be incorporated straightforwardly). Let $\Delta_m = |\log(P_{\text{prod}}) - \log(P_m)|$ denote the difference for reactant $m \in r_i$. The average difference is then

$$P_{\text{diff}} = \frac{1}{|r_i|} \sum_{m \in r_i} \Delta_m,\tag{23}$$

where $r_i$ denotes the set of reactants of reaction $R_i$.

We convert $P_{\text{diff}}$ into a separation cost using a threshold-based penalty function $C(P_{\text{diff}})$, where larger $\log(P)$ differences indicate easier separation and thus lower cost:

$$C_{\text{Scale}}(R_i) = C(P_{\text{diff}}) = \begin{cases} 0.0, & P_{\text{diff}} \geq 3.0 \quad \text{(excellent separation)} \\ 0.2, & 2.5 \leq P_{\text{diff}} < 3.0 \\ 0.4, & 2.0 \leq P_{\text{diff}} < 2.5 \\ 0.6, & 1.0 \leq P_{\text{diff}} < 2.0 \\ 0.8, & 0.5 \leq P_{\text{diff}} < 1.0 \\ 1.0, & P_{\text{diff}} < 0.5 \quad \text{(very poor separation)}. \end{cases}\tag{24}$$

To estimate future economic cost, we employ a heuristic analogous to the synthetic accessibility (SA) score, but based on predicted market prices. We use MolPrice (Hastedt et al., 2025), which captures both synthesizability and economic cost, to estimate the price of a molecule. The scale-up value of a molecule $m$ is defined as the normalized predicted price:

$$V_{m,\text{Scale}} = \frac{\text{MolPrice}(m)}{15}, \tag{25}$$

where the normalization constant is chosen based on empirical bounds, with 0 corresponding to the lowest and 15 to the highest observed prices.

**Guidance Objective**

The purpose of this objective is to guide the search towards available building blocks. The objective is inherited from existing multi-step algorithms. In particular, for all three single-step models investigated, the guidance cost is defined as:

$$C_{\text{Guid}} = \frac{-\ln P_{ret}(R_i)}{10}, \tag{26}$$

where $P_{ret}(R_i)$ is the likelihood of a reaction $R_i$ occurring, calculated by the single-step model. The cost is divided by 10 and then clipped to a range of $[0, 1]$.

The heuristic for NeuralSym is inherited from Chen et al. (2020); please refer to the paper for more information. For G2E and PDVN, the heuristic $V_{m,\text{Guid}}$ is set to zero. This is because the single-step models are trained with feedback from the environment via RL. This means that the resulting policy (the single-step model itself) is already aware of promising synthesis pathways, *i.e.*, those with all leaf nodes as building block molecules.

# D. Search Hyperparameters

Below we summarize the search hyperparameters used for the different weight-sampling strategies investigated in this work.

For all experiments, once a building block is reached, its future value estimate is set to $\mathbf{V}_m = \mathbf{0}$. The single-step retrosynthesis model returns $K = 25$ candidate reactions per molecule (apart from PDVN where $K = 50$), while the reaction condition predictor proposes two likely conditions per reaction.

## D.1. Bayesian Optimization.

The BO-based sampling strategy uses the following global parameters:

- Weight iteration frequency/budget ($w_{\text{budget}}$): 12

- Number of parallel weights per iteration ($N_S$): 5

**BO Strategy**

In addition to these global settings, BO incorporates a decay mechanism that reduces the influence of previously explored regions of the weight space. While a weight vector may initially yield large improvements, repeatedly sampling its local neighborhood is unlikely to further improve the Pareto front once that trade-off region has been sufficiently explored.

To account for this effect, we apply an age-dependent decay to the utility signal used to train the surrogate model. Let $\tau_j$ denote the number of BO iterations since weight $\mathbf{w}_j$ last produced a Pareto improvement. The decayed utility is defined as

$$u_j = \lambda^{\tau_j} \cdot u_j^{(0)}, \tag{27}$$

where $\lambda \in (0, 1)$ is a decay factor. If $\tau_j$ exceeds a maximum age $\tau_{\max}$, the utility is set to zero. The initial utility $u_j^{(0)}$ is computed as the hypervolume improvement induced by $\mathbf{w}_j$:

$$u_j^{(0)} = \Delta HV_j = HV_{\text{new}} - HV_{\text{old}}. \tag{28}$$

This formulation discourages repeated sampling in the vicinity of previously successful weights and promotes exploration of under-explored regions of the simplex.

To initialize the surrogate model with informative utility observations, we employ a warm-up phase based on deterministic grid sampling. During this phase, weight vectors $\mathbf{w}$ are drawn from a coarse grid over the $(N_D - 1)$-dimensional simplex:

$$\mathbf{w} \in \{0, 0.25, 0.5, 0.75, 1\}^{N_D} \quad \text{subject to} \quad \sum_{k=1}^{N_D} w_k = 1. \tag{29}$$

To bias early exploration toward feasible synthesis routes, we further restrict the initial weights by enforcing a minimum mass on the guidance-related objective. Specifically, we retain only grid points where the weight assigned to the guidance objective (the last dimension) satisfies $w_{N_D} \geq 0.5$ during the warm-up phase.

After the warm-up phase, candidate weight vectors are generated either from a refined grid or via Sobol sampling. These candidates define the search space for Bayesian optimization, from which the BO acquisition function selects the next batch of weights to evaluate.

**BO Hyperparameters.**

- Number of warm-up weights: 10

- Utility decay factor ($\lambda$): 0.5

- Maximum age ($\tau_{\max}$): 2

- Surrogate model: Gaussian Process with RBF kernel

- Kernel lengthscale: [0.05, 0.5]

**D.2. Grid Sampling**

Grid sampling is conceptually much simpler compared to BO. We sample weights from a coarse grid over the $N_D - 1$ simplex with resolution 0.33, i.e.,

$$\mathbf{w} \in \{0, \tfrac{1}{3}, \tfrac{2}{3}, 1\}^{N_D} \quad \text{subject to} \quad \sum_{k=1}^{N_D} w_k = 1. \tag{30}$$

The global parameters are:

- Weight iteration frequency/budget ($w_{\text{budget}}$): 16

- Number of parallel weights per iteration ($N_S$): 5

**D.3. Sobol (Random)**

Sobol sampling draws weights from the Sobol sequence. The global parameters are:

- Weight iteration frequency/budget ($w_{\text{budget}}$): 10

- Number of parallel weights per iteration ($N_S$): 5

- Total samples drawn from Sobol sequence: 32 (+ 4 extreme points)

# E. Supplementary Results

## E.1. Diverse Datasets

We also tested the G2E and NeuralSym models on the ChEMBL and Pistachio-reachable datasets, with the findings presented in the tables below.

*Table 5.* Summary of multi-objective performance on the ChEMBL and Pistachio-reachable datasets for three different single-step models. Our algorithm with BO sampling is compared to the Fixed and single-objective Retro* baselines.

| Model | Method | Pareto front Statistics | | MORetro* Dominance | |
|---|---|---|---|---|---|
| | | HV ($\uparrow$) | R2 ($\downarrow$) | Base. Dom. % ($\uparrow$) | Self Dom. % ($\downarrow$) |
| G2E (ChEMBL) | Retro* | $0.97 \pm 0.40$ | $0.49 \pm 0.14$ | 26% | 9% |
| | Fixed | $0.98 \pm 0.42$ | $0.49 \pm 0.12$ | 11% | 2% |
| | MORetro* (BO) | $\mathbf{1.02} \pm 0.38$*** | $\mathbf{0.45} \pm 0.14$*** | - | - |
| NeuralSym (Pistachio-reachable) | Retro* | $1.03 \pm 0.25$ | $0.48 \pm 0.12$ | 17% | 1% |
| | Fixed | $1.05 \pm 0.26$ | $0.46 \pm 0.12$ | 4% | 1% |
| | MORetro* (BO) | $\mathbf{1.08} \pm 0.22$*** | $\mathbf{0.44} \pm 0.14$*** | - | - |

Statistical significance MORetro* vs. baselines: *** $p < 8 \times 10^{-3}$ (Bonferroni), ** $p < 0.01$, *$p < 0.05$; Wilcoxon Test

*Table 6.* Success rate (finding at least 1 synthesis route per molecule) of MORetro* versus baselines on ChEMBL and Pistachio-reachable

| Model | Method | Success Rate (%) | $\Delta$Retro* % |
|---|---|---|---|
| G2E (ChEMBL) | Retro* | 93.3 | - |
| | Fixed | 86.0 | -7.3 |
| | MORetro* (BO) | 91.3 | -2.0 |
| NeuralSym (Pistachio-reachable) | Retro* | 98.6 | - |
| | Fixed | 97.9 | -0.7 |
| | MORetro* (BO) | 99.3 | +0.7 |

*Table 7.* Chemical diversity (div.) of Pareto-optimal synthesis routes. DS is the dissimilarity between a pair of routes.

| Model | Method | Max. DS ($\uparrow$) | Div. Frac. ($\uparrow$) |
|---|---|---|---|
| G2E | Retro* | $0.75 \pm 0.25$ | $0.39 \pm 0.24$ |
| | Fixed | $0.79 \pm 0.24$ | $0.46 \pm 0.25$ |
| | MORetro* (BO) | $\mathbf{0.82} \pm 0.23$** | $\mathbf{0.48} \pm 0.23$** |
| NeuralSym | Retro* | $0.74 \pm 0.29$ | $0.42 \pm 0.31$ |
| | Fixed | $0.82 \pm 0.25$ | $0.48 \pm 0.28$ |
| | MORetro* (BO) | $\mathbf{0.84} \pm 0.25$* | $\mathbf{0.51} \pm 0.27$** |

Statistical significance: ** $p < 0.01$, * $p < 0.05$; Wilcoxon Test

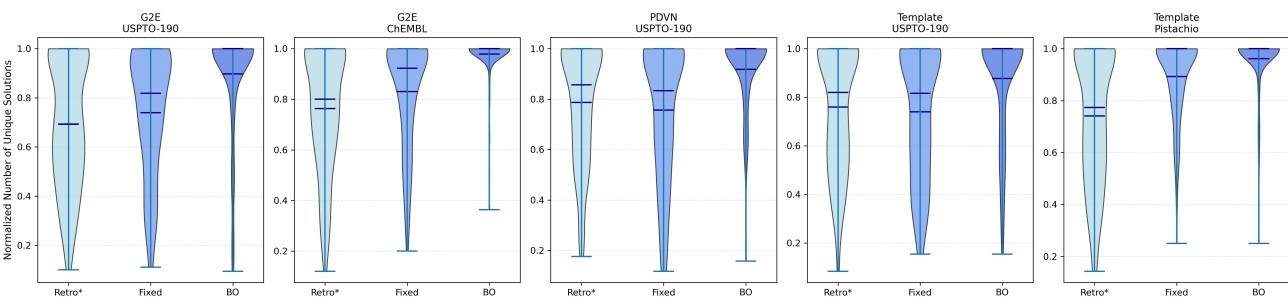

*Figure 6.* Per-molecule normalized number of unique solutions for all datasets investigated.

## E.2. Additional Experiments

### E.2.1. RESULTS FOR ADDITIONAL (FIXED) WEIGHTS

*Table 8.* Additional fixed-weight results on USPTO-190. Each weight vector **w** fixes the scalarization coefficients for the four objectives. HV is reported as mean $\pm$ std; superscripts denote significance vs MORetro$^*$ (BO) ($^{**}$: $p < 0.01$, $^*$: $p < 0.05$). "Baseline Dom." is the percentage of the weight's Pareto front dominated by MORetro$^*$ (BO); "Self Dom." is the percentage of the MORetro$^*$ (BO) front dominated by the weight.

| Model | Weights w | Success Rate (%) | HV (mean$\pm$std) | Baseline Dom. (%) | Self Dom. (%) |
|---|---|---|---|---|---|
| G2E | [0.25, 0.25, 0.25, 0.25] | 88.42 | $0.97 \pm 0.42^{**}$ | 20.07 | 3.10 |
| | [0.33, 0.33, 0.33, 0.01] | 90.53 | $1.00 \pm 0.38^{**}$ | 12.47 | 5.25 |
| | [1.00, 0.00, 0.00, 0.00] | 99.47 | $0.91 \pm 0.35^{**}$ | 37.33 | 7.72 |
| | [0.00, 1.00, 0.00, 0.00] | 97.37 | $0.90 \pm 0.40^{**}$ | 26.09 | 3.34 |
| | [0.00, 0.00, 1.00, 0.00] | 83.68 | $0.75 \pm 0.51^{**}$ | 42.69 | 1.57 |
| PDVN | [0.25, 0.25, 0.25, 0.25] | 65.26 | $0.71 \pm 0.55^{**}$ | 39.79 | 1.84 |
| | [0.33, 0.33, 0.33, 0.01] | 54.74 | $0.63 \pm 0.59^{**}$ | 47.75 | 2.31 |
| | [1.00, 0.00, 0.00, 0.00] | 67.37 | $0.56 \pm 0.51^{**}$ | 56.99 | 3.02 |
| | [0.00, 1.00, 0.00, 0.00] | 77.37 | $0.68 \pm 0.48^{**}$ | 39.71 | 2.43 |
| | [0.00, 0.00, 1.00, 0.00] | 15.79 | $0.15 \pm 0.38^{**}$ | 89.86 | 0.00 |
| NeuralSym | [0.25, 0.25, 0.25, 0.25] | 40.00 | $0.42 \pm 0.54^{**}$ | 36.89 | 2.53 |
| | [0.33, 0.33, 0.33, 0.01] | 36.84 | $0.40 \pm 0.54^{**}$ | 36.31 | 1.23 |
| | [1.00, 0.00, 0.00, 0.00] | 50.53 | $0.39 \pm 0.48^{**}$ | 32.30 | 18.97 |
| | [0.00, 1.00, 0.00, 0.00] | 54.74 | $0.44 \pm 0.52^{**}$ | 21.95 | 14.14 |
| | [0.00, 0.00, 1.00, 0.00] | 16.84 | $0.17 \pm 0.40^{**}$ | 73.67 | 0.00 |

### E.2.2. IMPACT OF SAMPLING STRATEGIES

Comparative study between different sampling strategies. Note that the reported R2 values for BO in Table 9 will differ from those reported in the main manuscript. This is because molecules that have no solutions are excluded for the R2 calculation across all methods. As we are comparing BO to different methods than in the main body, different molecules will be excluded.

*Table 9.* Comparison between BO, grid, and Sobol (Random) sampling in terms of Pareto front quality for all datasets

| Model | Dataset | Method | HV | R2 | # Solutions | Retro$^*$ | Fixed | Statistical |
|---|---|---|---|---|---|---|---|---|
| | | | mean$\pm$std | mean$\pm$std | mean$\pm$std | Dominated (%) | Dominated (%) | Significance |
| G2E | USPTO-190 | Grid | 1.03±0.32 | 0.42±0.11 | 13.70±10.99 | 30.9±34.7 | 19.3±34.7 | BO$^*$ >G$^{***}$ > R |
| | | Random | 1.01±0.37 | 0.45±0.10 | 11.57±9.41 | 29.8±34.2 | 15.2±30.0 | |
| | | BO | 1.04±0.31 | 0.42±0.11 | 13.45±10.08 | 34.0±36.2 | 21.0±35.8 | |
| | ChEMBL | Grid | 1.02±0.39 | 0.45±0.14 | 12.44±12.19 | 24.2±32.8 | 9.7±24.8 | G $\simeq$ BO $^{***}$ > R |
| | | Random | 1.01±0.39 | 0.46±0.14 | 11.19±11.20 | 23.0±32.5 | 8.7±23.6 | |
| | | BO | 1.02±0.38 | 0.45±0.14 | 12.24±11.78 | 24.0±33.2 | 9.7±24.9 | |
| PDVN | USPTO-190 | Grid | 0.90±0.41 | 0.44±0.12 | 12.12±11.14 | 18.6±30.3 | 11.2±26.2 | BO$^{***}$ > G$^{***}$ > R |
| | | Random | 0.86±0.43 | 0.46±0.11 | 10.18±9.11 | 15.8±28.6 | 8.8±23.4 | |
| | | BO | 0.93±0.39 | 0.44±0.11 | 12.30±10.76 | 19.6±31.0 | 11.9±26.8 | |
| NeuralS | USPTO-190 | Grid | 0.52±0.54 | 0.45±0.12 | 4.96±7.63 | 23.0±35.1 | 16.3±33.7 | BO$^{***}$ > G$^{***}$ > R |
| | | Random | 0.50±0.54 | 0.48±0.12 | 4.16±6.46 | 21.9±33.1 | 9.7±25.6 | |
| | | BO | 0.56±0.51 | 0.45±0.12 | 5.13±7.50 | 31.0±37.2 | 17.6±34.8 | |
| | Pistachio | Grid | 1.06±0.26 | 0.44±0.13 | 9.52±8.10 | 16.5±26.6 | 3.3±11.5 | G $\simeq$ BO $\simeq$ R |
| | | Random | 1.06±0.26 | 0.44±0.13 | 9.34±7.75 | 16.9±27.3 | 2.4±8.1 | |
| | | BO | 1.06±0.24 | 0.44±0.14 | 9.71±8.26 | 17.1±27.3 | 3.9±13.8 | |

The table below compares the success-rate of different sampling strategies on the investigated datasets.

*Table 10.* Comparison between BO, grid, and Sobol (Random) sampling in terms of success rate ($\downarrow$ failed targets)

| Model | Dataset | Grid | Random | BO | Total |
|---|---|---|---|---|---|
| G2E | USPTO-190 | 5 | 13 | **4** | 190 |
| | ChEMBL | **13** | 14 | **13** | 150 |
| PDVN | USPTO-190 | 11 | 18 | **6** | 190 |
| NeuralSym | USPTO-190 | 88 | 95 | **76** | 190 |
| | Pistachio | 2 | 3 | **1** | 150 |

### E.3. Comparison to MO-MCTS

#### E.3.1. MO-MCTS FOR ADDITIVE OBJECTIVES

Relative to the original implementation by Lai et al. (2025), the algorithmic steps (selection, expansion, rollout, and backpropagation) remain unchanged. The only methodological change is the reward definition, which is adapted for additive reaction costs.

In our implementation, the terminal branch value ($R_j(b)$) for objective $j$ is computed via:

$$\xi_j(b) = \text{length}(b) - k \sum_i \tilde{r}_j(R_i), \tag{31}$$

$$W_j(b) = \max\left(0, \frac{L_{\max} - \xi_j(b)}{L_{\max}}\right), \tag{32}$$

$$R_j(b) = z \cdot W_j(b), \tag{33}$$

where $L_{\max}$ is the search depth limit, $k$ is a scaling factor, and $z$ is a rollout-status multiplier. This follows the same strategy as proposed by Segler & Waller (2018).

Using our cost notation $C_j(R)$, we define the reward $\tilde{r}_j$ as:

$$\tilde{r}_j(R_i) = \begin{cases} 1 - C_j(R_i), & j \in \{\text{Sust}, \text{Tox}, \text{Scale}\}, \\ C_j(R_i), & j = \text{Guide}. \end{cases} \tag{34}$$

The status multiplier is

$$z = \begin{cases} > 1, & \text{if the rollout leaf is fully solved,} \\ \in [0, 1], & \text{if partially solved (solved-molecule ratio),} \\ -1, & \text{if no molecule is solved.} \end{cases} \tag{35}$$

All other components are identical to the original AiZynthFinder MO-MCTS (including edge statistics updates and child scoring/selection policy) (Lai et al., 2025).

#### E.3.2. PRELIMINARY RESULTS

Table 11 shows that our re-implementation of MO-MCTS is not competitive under the fixed budget of $N_B = 300$. We believe the cause to be twofold: 1) Rollouts drain the single-step budget quickly without guaranteeing to discover new synthesis routes. 2) Rollouts already suffer from high variance when optimizing one objective. This is exacerbated when one tries to estimate $N_D$ objectives. Note that no hyperparameter tuning was performed for MO-MCTS.

*Table 11.* Comparison to re-implementation of MO-MCTS

| Model | Success rate (%) | HV ($\uparrow$) - all mols | HV ($\uparrow$) - only successful mols | $\Delta HV$ vs. MORetro* |
|---|---|---|---|---|
| G2E | 95.3 | $0.43 \pm 0.48$ | $0.45 \pm 0.48$ | -0.61 |
| PDVN | 93.7 | $0.60 \pm 0.44$ | $0.64 \pm 0.42$ | -0.32 |
| NeuralSym | 33.7 | $0.17 \pm 0.37$ | $0.51 \pm 0.48$ | -0.39 |

## F. Toxicity Assessment Prompt Template

---

**Toxicity Assessment Prompt Template (Excerpt)**

**System role:** Expert toxicologist and chemist.

**Task:** Given a molecular structure in SMILES notation, assign a toxicity score in the range $[0, 1]$.

**Toxicity scale:**

- 0.0 - Non-toxic

- $0.1 - 0.3$ - Low toxicity

- $0.4 - 0.6$ - Moderate toxicity

- $0.7 - 0.9$ - High toxicity

- 1.0 - Extremely toxic

**Evaluation procedure (priority order):**

1. Check custom reference studies for exact or similar compounds

2. Acute toxicity data (e.g. $LD_{50}$, $LC_{50}$)

3. Environmental impact and persistence

4. Regulatory hazard classifications (GHS, OSHA)

5. Structural alerts and known mechanisms of toxicity

**Special rule for transition-metal catalysts:** If a transition metal appears in the SMILES, a metal-specific baseline toxicity score from custom catalyst greenness data is used and adjusted based on ligand effects (e.g. chelation or increased bioavailability).

**Output format:**

```
Score: [0—1]
Explanation: 2—3 sentence justification citing reference data when
available
```

---

