# OpenReview forum: "From Feasible to Practical: Pareto-Optimal Synthesis Planning"
_ICML.cc/2026/Conference — ICML 2026 spotlight_

### Official Review · Reviewer_M3Da · 2026-03-05

**Soundness:** 3
**Presentation:** 3
**Significance:** 4
**Originality:** 3
**Overall Recommendation:** 5
**Confidence:** 3

**Summary:**

The paper proposes MORetro*, a multi-objective A* search algorithm for synthesis planning that  captures trade-offs among objectives like sustainability, toxicity, and cost. It uses weight-based scalarization and Bayesian Optimization sampling while deriving theoretical bounds that guarantee Pareto optimality for fixed models. Experiments show that MORetro* outperforms single-objective baselines in Pareto front quality and route diversity while preserving the success rate.

**Compliance With Llm Reviewing Policy:**

Affirmed.

**Final Justification:**

From a chemical engineering perspective, the study's significance is excellent, particularly due to its  industry aware objectives such as sustainability, scalability, and toxicity. The authors demonstrate a clear effort to estimate these metrics detailed in appendix and provide explanations, along with outlining future work to address my concerns. I maintain my original score, with revised significance and confidence.

**Key Questions For Authors:**

1. How does the batching impact the wall-clock inference time compared to single-objective baselines?
2. The definition, prediction/approximation accuracy, and uncertainty in the metrics (sustainability, toxicity, and scale-up potential) can significantly impact the final results. In particular, this is important when the variance in result statistics is high, while the difference compared to baseline is small in table 1 and 2.  Also, these metrics are only defined and explained in a simple way.

**Limitations:**

The authors briefly mention the success rate gap in the conclusion.

**Strengths And Weaknesses:**

Soundness: The study is backed by theoretical bounds for Pareto-optimality and a experimental design across multiple single-step models.

Presentation: The study is clearly structured, with effective visual representations of the search graph and pareto fronts. One minor issue is that important details regarding objective cost functions and toxicity model prompts are in the appendices due to space limitation.

Significance: This study aligns synthesis planning to multi-objective industrial decision-making process. By modeling trade-offs like toxicity and cost, it advances the application of ML in chemistry.

Originality: The study combines multi-objective A* search, linear scalarization, and Bayesian optimization within retrosynthesis. It introduces theoretical guarantees for safe pruning that distinguish it from other approaches.

Weakness:

- The metrics (sustainability, toxicity, and scale-up potential) are approximated via relatively simple ways. The inherent  impacts of uncertainty, error, bias in these prediction models on the final outcome are not thoroughly discussed. This could result in strong variation of the main results.

---

> ### Author Rebuttal · Authors · 2026-03-30
>
> We thank the reviewer for their feedback! Please find our answers to their key concerns below:
>
> **Inference time**
>
> The computationally heaviest part of the entire pipeline is the QUARC condition model, as it needs to be called $k$ times given that the single-step predictor returns $k$ reactions. Our algorithm supports batching for both the single-objective and multi-objective models: the single-objective baseline processes $k$ reactions per target per iteration, whereas our algorithm processes up to $N_S \cdot k$ reactions across $N_S$ targets simultaneously. In our implementation, the single-objective and multi-objective algorithms therefore have the same inference time per iteration.
>
> However, if one did not use batching for predicting the reaction conditions, the following difference in inference time would be found (tested on an NVIDIA RTX6000) for the NeuralSym and G2E models:
>
> 1. NeuralSym + QUARC - With batching: 0.70 sec/mol; Without batching: 0.92 sec/mol (30\% speed-up)
> 2. G2E + QUARC - With batching: 1.9 sec/mol; Without batching: 2.3 sec/mol (19\% speed-up)
>
> The G2E model is expensive due to slow CPU-bound operations via RDKit which cannot be efficiently parallelized.
>
> **Uncertainty in Objectives**
>
> It is true that the recovered Pareto front is highly dependent on the prediction accuracy of the objectives. The focus of the paper is however not on the reliability of the reaction/condition predictor or the objective surrogate models, but rather on how to uncover Pareto-optimal synthesis pathways given any objectives and any predictor. As the field advances, we expect the accuracy of these surrogate models to improve over time.
>
> We nonetheless acknowledge the reviewer's concerns regarding variance in these objectives. We delegate this to future work in the added limitation and outlook section as follows (see last sentence):
>
> *MORetro$^\ast$ assumes that reaction costs are additive, currently limiting objectives to those that decompose naturally per reaction or molecule. A principled extension could use admissible additive lower bounds to guide search while evaluating the true non-additive cost only upon route completion, preserving theoretical guarantees via pruning. However, the practical heuristics currently employed are not provably admissible, meaning the pruning bound falls back to the zero vector in practice, reducing pruning efficiency. Constructing tight admissible lower bounds for such objectives remains an open challenge, which we leave for future work. Furthermore, the current study assumes a fixed number of 3 Pareto objectives. Future work will investigate the scalability of our algorithm to a higher number of diverse objectives. While out of scope for this study, future work will address the robustness and accuracy of the selected objectives to further push adoption of CASP within industry.*

---

> > ### Author Rebuttal · Reviewer_M3Da · 2026-04-03
> >
> > Thank you for the detailed response. I appreciate the clarifications provided, which have addressed my concerns. I will maintain my positive recommendation.

---

### Official Review · Reviewer_jq3D · 2026-03-08

**Soundness:** 4
**Presentation:** 4
**Significance:** 4
**Originality:** 3
**Overall Recommendation:** 5
**Confidence:** 4

**Summary:**

This paper addresses a key gap in computer-aided synthesis planning (CASP) by formulating retrosynthesis as a multi-objective search problem. The authors propose MORetro*, an extension of Retro* that uses weight-sum scalarization with principled weight sampling (including a BO-based strategy) to generate Pareto fronts over user-defined objectives such as sustainability, toxicity, and scale-up potential; they further derive a vector-valued lower bound and a bound-dominance termination/pruning condition that provides a completeness guarantee for recovering the exact Pareto front given admissible heuristics. Empirically, on USPTO-190 (and additional datasets in the appendix) and across three distinct single-step models, MORetro* consistently improves hypervolume and diversity relative to single-objective and fixed-scalarization baselines, while maintaining near-baseline route-finding success rates.

**Compliance With Llm Reviewing Policy:**

Affirmed.

**Final Justification:**

I am keeping the score at a 5, rather than raising it to a 6 (Strong Accept), for two reasons:

1. Algorithmic Scope: The method is a masterful integration of MOO, BO, and A* search atop the Retro* framework. However, for a venue like ICML, the contribution leans more toward an excellent application of existing techniques tailored for chemistry with insightful integrations/modifications, rather than a fundamental breakthrough in core ML search paradigms.

2. Heuristic Limitations: While the theoretical guarantees (Theorem 3.4) are sound, their practical efficacy is heavily constrained by the noisy proxy metrics used for estimating future costs (), such as SA scores and EToxPred. Relying on these imperfect estimators inherently caps the real-world reliability of the generated Pareto fronts.

Overall, this is a strong, well-executed application paper that effectively bridges theoretical route-finding with practical constraints. I firmly support its acceptance.

**Key Questions For Authors:**

Are the proposed heuristics (SA-based, EToxPred-based, MolPrice-based) guaranteed to be admissible lower bounds for the respective cumulative future costs? If not, do you cap them to preserve admissibility in the implementation (e.g., by setting Vm = 0 for pruning but still using the heuristics for selection)?
Can you provide an ablation quantifying the effect of the V_bound pruning on (i) expansions, (ii) runtime, and (iii) final PF quality under the same budget?
1. How are multiple condition predictions per reaction handled in the graph (e.g., branching into separate AND nodes per condition set)? 2. Does this significantly increase branching factor, and how sensitive are results to the number of condition proposals?
3. Please clarify the “guiding convergence objective”: is it included in the scalarization for selection for all methods, and is it excluded from PF metrics for all methods (including Fixed)? Could this asymmetry advantage/disadvantage certain baselines?
4. Could you add a stronger multi-objective baseline by running Retro* repeatedly with many random weights (union of solutions), matched for total single-step calls/time, and compare PF quality/diversity?
5. What are the wall-clock runtimes and hardware used for all methods, and how do BO vs. grid/Sobol sampling compare in time overhead per iteration?
6. How sensitive are results to the number of objectives (e.g., 4–5 objectives)? Do you observe degradation in success rate or BO sampling stability as dimensionality increases?
7. Regarding agent toxicity scores: will you release the curated mapping (including the LLM-labeled portion) to ensure reproducibility?

**Limitations:**

See weaknesses

**Strengths And Weaknesses:**

## Strengthes
1. ** Technical novelty and innovation**
* Introduces a multi-objective extension of Retro* that conducts parallel, scalarized A*-style searches under multiple weights and shares expansions on a common AND-OR graph.
* Proposes a vector-valued “pruning number” and V_bound that yield a bound-dominance termination criterion; the resulting theorem provides a clear completeness guarantee for recovering the full Pareto set when admissible lower bounds are available.
* Integrates a reaction condition predictor during search, enabling optimization of practically relevant objectives that require conditions (temperature, catalysts/solvents); this is a substantive step towards practice-aligned CASP.
* Solution-informed BO sampling (GIBBON-based batch acquisition with a decayed utility) is a sensible and effective way to adaptively explore the weight simplex.
2. **Experimental rigor and validation**
* Evaluates across three qualitatively different single-step models (template-free and template-based), showing algorithmic independence from the black-box one-step predictor.
* Uses standard multi-objective indicators (hypervolume and R2), dominance coverage measures, and a chemistry-aware diversity metric; statistical testing (Wilcoxon + Bonferroni) is applied.
* Reports success rates comparable to single-objective Retro* to demonstrate MORetro* largely preserves solvability.
3. **Clarity of presentation**
* The method is clearly explained with a compact algorithm box and a helpful flow diagram; the propagation/update rules and weight-scalarized selection steps are explicit.
* Theoretical results are stated with definitions and an accessible proof sketch; the role of admissibility is made explicit.
4. **Significance of contributions**
* Multi-objective route planning is directly aligned with real-world decision-making in pharma/materials (e.g., balancing greenness, cost, toxicity, and scalability).
* The framework provides a general recipe to turn existing single-objective planners into multi-objective ones with principled guarantees, and empirically demonstrates meaningful improvements in front quality and diversity.

## Weaknesses
1. **Technical limitations or concerns**
* The completeness guarantee hinges on admissible lower bounds Vm; while a zero vector is always admissible (remark), the practical heuristics proposed (SA score, EToxPred, MolPrice) are not shown to be provably admissible lower bounds for the cumulative downstream costs as defined—this weakens the practical force of the guarantee.
* Linear scalarization is known not to recover unsupported Pareto points in non-convex/frontier regions; the paper’s guarantee avoids this limitation only under the (strong) termination-by-bound-dominance setting, which is not used in the time/budget-limited experiments.
2. **Experimental gaps or methodological issues**
* Baselines focus on single-objective Retro* and a single fixed-weight scalarization; a stronger multi-objective baseline would be to run multiple independent scalarized Retro* searches under many random weights (union of routes) and compare fronts and budgets on equal footing.
* There is no ablation quantifying the impact of the proposed pruning bound (V_bound-based pruning) on runtime/expansions or front quality.
* Runtime and wall-clock compute are not reported, making it hard to judge overheads from parallel scalarizations and BO vs. grid/Sobol sampling.
* The method is claimed to handle “arbitrary number of objectives,” but experiments are restricted to three (plus a guiding convergence objective that is excluded from the Pareto analysis); scalability to higher dimensions is not assessed.
3. **Clarity or presentation issues**
* The role of the “guiding” convergence objective (excluded from PF computation) vis-à-vis baselines could be more precisely specified to avoid confounding; the Fixed baseline appears to allocate weight to convergence, although that dimension is excluded from PF scoring—this should be clarified as it may bias search behavior differently across methods.
* Details on how multiple condition predictions per reaction (two per QUARC) are integrated into branching and costs would aid reproducibility.
4. **Missing related work or comparisons**
* The paper cites MO-MCTS in retrosynthesis but omits broader multi-objective search/shortest-path literature (e.g., Martins’ multiobjective shortest path, MOA* variants in planning/graphs, label-setting approaches or other multiobjective algorithems) which would help contextualize the proposed bound and selection strategy.
* A comparison to simple Pareto-aware search policies (e.g., random hypervolume scalarizations, ParEGO-style strategies, or EHVI-inspired weight schedules) would be informative given the BO component.

---

> ### Author Rebuttal · Authors · 2026-03-30
>
> We thank the reviewer for their detailed and insightful comments, which we address below. Due to the character limit, we are constrained to refer you to our comments to the other reviewers in some cases. We apologize for any resulting inconvenience.
>
> **Ablation study on pruning via $\mathbf{V}_{bound}$**
>
> It is correct that the practical heuristics cannot be used for pruning as they are not admissible lower bounds. We performed the suggested ablation study, activating the pruning bound during search (Table 1 in the paper does not use pruning for a fair comparison). We found that the pruning bound improves Pareto front quality while also reducing the search space on average by 50\% and decreasing the average expansion budget (as 11 molecules were found to be Pareto-optimal). For the detailed findings, please refer to our comments to Reviewer w6z6 under **Ablation on $V_{bound}$**.
>
> We would like to point out that the runtime and expansion budget can only be reduced if molecules are found to be Pareto-optimal; otherwise the algorithm continues until the expansion budget is exhausted.
>
> **Handling of multiple conditions**
>
> Multiple conditions do not increase the branching factor of our algorithm during search (we build a graph as in the RetroGraph paper, not a tree). Each new condition is spawned as a separate reaction node, whose children reactant nodes are connected independently. Crucially, reactant nodes are not duplicated; reaction nodes share the same parent and children. This is possible thanks to the scalarization of objectives: the best reaction node is simply the one with the lowest scalar value, allowing cost propagation without ambiguity.
>
> The only challenge arises during solution recording. Keeping track of all solutions in an $N$-dimensional space would lead to a combinatorial explosion given our graph structure. Instead, we track only Pareto-optimal solutions via bottom-up propagation, circumventing this issue.
>
> **Role of convergence objective**
>
> The guiding objective is included in all scalarizations and excluded from the Pareto front metrics. It is reasonable to assume that this biases certain baselines. To address this concern, we ran experiments with different weight scalarizations and found that MORetro* outperforms the baselines in all cases. The exact results are shown under **Baselines for comparison** in our replies to Reviewer txVJ.
>
> **Stronger baseline comparison**
>
> We thank the reviewer for this suggestion; however, we question whether this would constitute a fair comparison. Running $N$ different weight vectors, each for 300 expansions, would result in a total of $N \times 300$ single-step model calls, significantly more compute than MORetro*, which uses 300 expansions in total. Alternatively, splitting 300 expansions across $N$ independent Retro* instances would give each instance only $300/N$ calls, which is unlikely to yield meaningful search graphs for large $N$ - making this an increasingly inferior baseline as $N$ grows. This highlights a fundamental advantage of MORetro*: by sharing a single AND-OR graph across all weight scalarizations, every expansion benefits all weight
> vectors simultaneously. If the instances were to share the same search graph, this would instead resemble our Sobol sampling baseline, which is already included in our comparisons.
>
> **Wall-clock runtimes**
>
> All experiments were run on the following hardware: NVIDIA RTX6000, 6 CPU cores (AMD EPYC 7742). On average, a target product completes search (300 expansions) in 6-12 minutes, depending on the search graph size and which single-step predictor is used. BO and grid/Sobol sampling have the same runtime, as both use 5 weights in parallel and thus benefit equally from batching. The effect of batching on the predictive pipeline is as follows, with the condition and single-step predictor taking more than 95% of inference time of the pipeline (due to thousands of expensive RDKit operations per molecule and the ML component):
>
> 1. NeuralSym + QUARC - With batching: 0.70 sec/mol; Without batching: 0.92 sec/mol (30\% speed-up)
> 2. G2E + QUARC - With batching: 1.9 sec/mol; Without batching: 2.3 sec/mol (19\% speed-up)
>
> **Scalability to a higher number of objectives**
>
> We appreciate the reviewer's concern and agree that performance may degrade as more objectives are added. We hypothesize that the sampling stability of the BO algorithm would deteriorate without increasing the warm-up period for exploring different weight combinations. Scalability and robustness of objectives will be investigated in future work (see **Limitations of objectives** in our response to Reviewer w6z6).
>
> **Agent toxicity scores**
>
> The agent toxicity scores are included in the code repository (moretro/external/tox\_scores/agents\_with\_scores.json). To improve visibility, we will upload this file to Figshare to avoid confusion.
>
> **Missing related work**
>
> We have added relevant related work (MOA* and variants) to Section 2.1 as suggested.

---

> > ### Author Rebuttal · Reviewer_jq3D · 2026-04-02
> >
> > Thank you for the thorough rebuttal, which successfully addressed my initial concerns. I am confidently maintaining my score of 5 (Accept) for this highly practical contribution.
> >
> > I am keeping the score at a 5, rather than raising it to a 6 (Strong Accept), for two reasons:
> >
> > 1. Algorithmic Scope: The method is a masterful integration of MOO, BO, and A* search atop the Retro* framework. However, for a venue like ICML, the contribution leans more toward an excellent application of existing techniques tailored for chemistry with insightful integrations/modifications, rather than a fundamental breakthrough in core ML search paradigms.
> >
> > 2. Heuristic Limitations: While the theoretical guarantees (Theorem 3.4) are sound, their practical efficacy is heavily constrained by the noisy proxy metrics used for estimating future costs ($V_m$), such as SA scores and EToxPred. Relying on these imperfect estimators inherently caps the real-world reliability of the generated Pareto fronts.
> >
> > Overall, this is a strong, well-executed application paper that effectively bridges theoretical route-finding with practical constraints. I firmly support its acceptance.

---

### Official Review · Reviewer_txVJ · 2026-03-12

**Soundness:** 4
**Presentation:** 4
**Significance:** 3
**Originality:** 3
**Overall Recommendation:** 6
**Confidence:** 4

**Summary:**

This paper introduces MORetro*, a multi-objective A* search framework designed for retrosynthesis. The method generalizes the Retro* algorithm by representing a molecule’s value/cost V(m) as a vector rather than a scalar. It expands the synthesis tree at nodes that maximize the scalarized value $V(m)w_j^T$ across different weight vectors $w_j$ that are sampled to maximize expected information gain. The authors provide theoretical guarantees that their method can recover the Pareto front for a fixed single-step model. Their evaluation demonstrates that MORetro* achieves superior performance in terms of convergence to ideal costs, Pareto coverage, and solution diversity, with a marginal decrease in overall success rate.

**Compliance With Llm Reviewing Policy:**

Affirmed.

**Final Justification:**

The authors have effectively addressed all of my concerns. Given the exceptional quality of the revised manuscript, I am pleased to raise my score to a 6.

**Key Questions For Authors:**

See weaknesses

**Limitations:**

no limitation section

**Strengths And Weaknesses:**

### Strengths

- Multi-objective retrosynthesis is a significant real-world challenge.
- The proposed method is both elegant and technically robust.
- The approach inherently improves the diversity of the generated synthesis pathways.
- The paper is clearly written, well-structured, and easy to follow.

### Weaknesses

- The manuscript lacks a dedicated limitations section.
- There is no comparison to Monte Carlo Tree Search (MCTS) methods.
- It is unclear which specific objective was used for the Retro* baseline. To ensure a fair comparison, I believe results for all four objectives should be reported.
- The rationale behind the choice of the preference vector for Fixed Retro* is not sufficiently explained. An extended study involving a wider range of preference vectors would ensure the results are robust and not cherry-picked.
- The practical implementation of the pruning mechanism remains unclear. Specifically, how is $V_m$  defined in Equation 16? If it refers to the $V_m$  introduced in Section 3.1, the heuristics used there do not appear to be admissible, which would impact the validity of the pruning. Is $V_m$  set to $0$ in practice as suggested in appendix B.1.? If so, please state this explicitly and provide further clarification on this section.
- The framework assumes that synthesis pathway costs are additive, which is not always the case. It would be helpful if the authors addressed whether this framework could be feasibly extended to non-additive cost formulations.

I believe this paper represents a strong contribution to the field, and I recommend acceptance. If my concerns are addressed, I would consider raising my score to a 6.

---

> ### Author Rebuttal · Authors · 2026-03-30
>
> We thank the reviewer for the positive feedback and helpful comments. We believe that we were able to further strengthen the paper thanks to their insights.
>
> **Limitations of current work**
>
> We have added a dedicated limitations & outlook section to the paper, which can be found in our replies to Reviewer w6z6 under (**Limitations of objectives**).
>
> **Comparison to MO-MCTS**
>
> We re-implemented the MO-MCTS by Lai et. al. with our objectives. To do so, we had to make assumptions regarding how to calculate the overall route reward that are directly coupled to the backpropagation and selection routines (*Lai et. al.* use terminal rewards). The results show our implementation of MO-MCTS is not competitive under the fixed budget. We believe this to be because of the high variance of rollouts; a known issue for single-objective planners, exacerbated when accurately estimating $N$ objectives. Additionally, each rollout uses about 15 single-step calls, consuming the expansion budget at a much faster rate. As we did not tune hyperparameters, we do not believe that comparing to MO-MCTS is fair. Nonetheless, we added an additional section to Appendix E - Supplementary Results detailing our implementation of MO-MCTS and including the table for the USPTO-190 dataset as shown below:
>
> | Model        | Success rate (\%)          |         HV ($\uparrow$) - all mols  |         HV ($\uparrow$) - only successful mols  |         $\Delta HV$ vs. MORetro$^\ast$  |
> |--------------|----------------------------|-------------------------------------|------------------------------------------------|-----------------------------------------|
> | G2E          | 95.3                       | $0.43 \pm 0.48$                     | $0.45 \pm 0.48$                                | -0.61                                   |
> | PDVN         | 93.7                       | $0.60 \pm 0.44$                     | $0.64 \pm 0.42$                                | -0.32                                   |
> | NeuralSym    | 33.7                       | $0.17 \pm 0.37$                     | $0.51 \pm 0.48$                                | -0.39                                   |
>
>
> **Baselines for comparison**
>
> The Retro* baseline was run using only the convergence objective ($\mathbf{w}=[0,0,0,1]$). To address concerns regarding the baselines, we ran additional experiments for several weight vectors as shown below. Due to computational constraints, we currently report results for the G2E and PDVN models on USPTO-190, with NeuralSym experiments ongoing as well as the final PDVN experiment. Overall, MORetro* (BO) clearly outperforms all weight combinations for both models with high statistical support.
>
> G2E:
>
> | Weights | Success Rate % | HV (mean ± std) | R2 (mean ± std) | Baseline Dominated % | Self Dominated % |
> |---|---|---|---|---|---|
> | [0.25, 0.25, 0.25, 0.25] | 88.42 | 0.97 ± 0.42*** | 0.47 ± 0.09*** | 20.07 | 3.10 |
> | [0.33, 0.33, 0.33, 0.01] | 90.53 | 1.00 ± 0.38*** | 0.47 ± 0.08*** | 12.47 | 5.25 |
> | [1.0, 0.0, 0.0, 0.0] | 99.47 | 0.91 ± 0.35*** | 0.48 ± 0.11*** | 37.33 | 7.72 |
> | [0.0, 1.0, 0.0, 0.0] | 97.37 | 0.90 ± 0.40*** | 0.46 ± 0.12*** | 26.09 | 3.34 |
> | [0.0, 0.0, 1.0, 0.0] | 83.68 | 0.75 ± 0.51*** | 0.44 ± 0.13 | 42.69 | 1.57 |
>
> PDVN:
>
> | Weights | Success Rate % | HV (mean ± std) | R2 (mean ± std) | Baseline Dominated % | Self Dominated % |
> |---|---|---|---|---|---|
> | [0.25, 0.25, 0.25, 0.25] | 65.26 | 0.71 ± 0.55** | 0.52 ± 0.08** | 39.79 | 1.84 |
> | [0.33, 0.33, 0.33, 0.01] | 54.74 | 0.63 ± 0.59** | 0.54 ± 0.07** | 47.75 | 2.31 |
> | [1.0, 0.0, 0.0, 0.0] | 67.37 | 0.56 ± 0.51** | 0.48 ± 0.12 | 56.99 | 3.02 |
> | [0.0, 1.0, 0.0, 0.0] | 77.37 | 0.68 ± 0.48** | 0.47 ± 0.13 | 39.71 | 2.43 |
>
> **Pruning via $\mathbf{V}_m$**
>
> You are correct that in practice we use $\mathbf{V}_m=0$, as the current heuristics are not admissible. To quantify the impact of pruning on the search, we performed an ablation study, which can be found in our replies to Reviewer w6z6 under **Ablation on $V_{bound}$**. We now also state explicitly in the main body (following Remark 3.6):
>
> *Since the heuristics defined in Section 3.4 are not admissible, we use the zero vector $\mathbf{V}_m = [0,...,0] \in R^N$ as a lower bound to calculate $\mathbf{V}$(bound).*
>
> **Additivity of costs**
>
> We agree that our framework currently only supports additive costs. To extend the framework to route-level objectives, one could investigate admissible reaction-level costs that provide a lower bound to the route-level cost. We delegate this to future work as a promising research direction (see reply to Reviewer w6z6).

---

> > ### Author Rebuttal · Reviewer_txVJ · 2026-04-03
> >
> > The authors have effectively addressed all of my concerns. Given the exceptional quality of the revised manuscript, I am pleased to raise my score to a 6.

---

### Official Review · Reviewer_w6z6 · 2026-03-13

**Soundness:** 3
**Presentation:** 3
**Significance:** 3
**Originality:** 3
**Overall Recommendation:** 5
**Confidence:** 4

**Summary:**

The authors present MORetro*, a multi-objective retrosynthesis search algorithm that generates Pareto-optimal synthesis routes given an arbitrary number of criteria. MORetro* effectively explores the Pareto front via several weight-sampling strategies and extends its single-objective predecessor Retro* by down-sampling the high-dimensional objective space via linear scalarization. The authors additionally derive a cost bound for synthesis routes, under which unpromising molecules can be pruned safely.

**Compliance With Llm Reviewing Policy:**

Affirmed.

**Key Questions For Authors:**

1. In the single-objective Retro* baseline, what exactly is the objective? Is it the reaction likelihood and cost heuristics that are inherited from the Retro* paper?
2. The sampled weight $w_j$ is applied to both the specific reaction cost $c(R)$ and the graph-independent heuristic $V_m$, even though they estimate fundamentally different quantities. Are their respective definitions as described in Appendix C sufficiently similar to justify this choice?

**Limitations:**

Yes

**Strengths And Weaknesses:**

**Strengths**
1. When benchmarked on the USPTO-190 dataset, MORetro* outperforms Retro* on Pareto front hypervolume when used with G2E, PDVN, and NeuralSym, demonstrating that the improvements to the algorithm are consistent and model-agnostic even between template-based and template-free predictors.
2. The MORetro* algorithm naturally permits Bayesian Optimization (BO) over the choice of weights $w$, guided by hypervolume improvements and diversity. Tables 1, 2 and Figure 4 indicate that this contributes to more efficient exploration of the Pareto front defined by toxicity, scale-up potential, and sustainability.

**Weaknesses**
1. MORetro* produces marginally lower success rates compared to Retro* on USPTO-190.
2. Although the paper provides theoretical justification for safe pruning via $V_{bound}$, no ablation is provided to indicate how performance improves in practice with respect to search space reduction. It may be worthwhile to quantify empirical improvements.
3. The AND/OR graph model for retrosynthesis makes the somewhat limiting assumption that the optimal synthesis pathway can be recovered by independently minimizing costs at each molecular node and summing them. Toxicity of intermediates and precursors generated along the route cannot be accurately decomposed per-reaction. This limits the authors to using EToxPred as a heuristic (Vm,Tox) to penalize molecules with toxic moieties and, in general, limits the scope of objectives to those that can satisfy this assumption.

---

> ### Author Rebuttal · Authors · 2026-03-30
>
> We thank the reviewer for their encouraging and insightful feedback, which helped us to improve the quality of our paper. Please find the answers to your key concerns below:
>
> **Single-objective baseline**
>
> The cost heuristic is directly inherited from the Retro* paper. The reaction likelihood is provided by the single-step model, which was taken from *Double-Ended Synthesis Planning with Goal-Constrained Bidirectional Search*, Yu et. al. (2024).
>
> **Ablation on $V_{bound}$**
>
> We ran additional experiments to quantify performance improvement and pruning signal in the search graph using $V_{bound}$, as well as using $V_{bound} + \epsilon$ (with $\epsilon=0.1$) to construct the $\epsilon$-Pareto Front for the *G2E* model on the USPTO-190 dataset. Below, we present the results and how it compares to MORetro* (BO) without using $V_{bound}$.
>
> 1. For $V_{bound}$, we found that the search space is on average reduced by $54 \pm 33$% (given an average number of 2600 open nodes upon termination, this means 1300 nodes are pruned on average). We also observed that for 11/190 molecules the optimal/true Pareto front was recovered. In total, 17/190 and 39/190 molecules had their search space reduced by at least 95\% and 90\% upon termination, respectively. In comparison to the MORetro* (BO) baseline, we see that for 82/190 molecules, the recovered Pareto front did not change. Nevertheless, we still observe significant statistical improvements in R2 value (0.41 vs 0.44 for MORetro* (BO)) overall with comparable HV (1.045 vs 1.040), indicating that the bound also leads to better Pareto fronts under the same expansion budget.
>
> 2. For $V_{bound} + \epsilon$, we observe that the Pareto front quality even slightly increases, essentially uncovering the same Pareto Front as the baseline. However, the algorithm now guarantees 32/190 molecules to be $\epsilon$-Pareto optimal, with 40/190 and 65/190 molecules having their search space reduced by 95\% and 90\% respectively. On average, the search space is reduced by $61 \pm 35$% upon termination. These results show that one can use the $\epsilon$ bound to make pruning more aggressive whilst still recovering a high quality Pareto Front.
>
> Please find the detailed results in the Table below, which is added to the manuscript in the Results section under **Ablation Studies**.
>
> | Methods | PO (100% cut) | 95% cut | 90% cut | Avg Reduction (%) | HV (↑) | R2 (↓) | Model calls |
> |---|---|---|---|---|---|---|---|
> | V_bound | 11 (6%) | 17 (9%) | 39 (21%) | 54 ± 33 | 1.045* (+0.005) | 0.41*** (-0.03) | 289 ± 50 |
> | ε-V_bound | 32 (17%) | 40 (21%) | 65 (34%) | 61 ± 35 | 1.045* (+0.005) | 0.43* (-0.01) | 265 ± 83 |
>
> **Limitations of objectives**
>
> Thanks for pointing out the current limitation wrt. the additivity of objectives. We added a dedicated limitation section to the manuscript which reads as follows:
>
> *MORetro$^\ast$ assumes that reaction costs are additive, currently limiting objectives to those that decompose naturally per reaction or molecule. A principled extension could use admissible additive lower bounds to guide search while evaluating the true non-additive cost only upon route completion, preserving theoretical guarantees via pruning. However, the practical heuristics currently employed are not provably admissible, meaning the pruning bound falls back to the zero vector in practice, reducing pruning efficiency. Constructing tight admissible lower bounds for such objectives remains an open challenge, which we leave for future work. Furthermore, the current study assumes a fixed number of 3 Pareto objectives. Future work will investigate the scalability of our algorithm to a higher number of diverse objectives. While out of scope for this study, future work will address the robustness and accuracy of the selected objectives to further push adoption of CASP within industry.*
>
> **Scalarization of $\mathbf{V}_m$**
>
> The choice of using the same scalarization for $\mathbf{V}_m$ and $\mathbf{c}(R)$ is justified at the level of the high-level objective rather than the specific measurement. For example, for industrial potential, the reaction cost captures mixture separability while the heuristic estimates downstream molecular price - two distinct quantities that are nevertheless both meaningful proxies for overall route viability at industrial scale. Applying the same weight vector is therefore consistent in the sense that it reflects a fixed preference over high-level objectives, with both the reaction cost and heuristic contributing complementary evidence toward the same goal. We acknowledge that this design choice implies the scalarization operates over heterogeneous quantities, and that a more principled approach could involve objective-specific weighting of costs and heuristics separately. We leave a more principled treatment of this to future work.

---

> > ### Author Rebuttal · Reviewer_w6z6 · 2026-04-03
> >
> > Thank you for elaborating on these points. I maintain my positive score and my recommendation to accept.

---

### Decision · Program_Chairs · 2026-04-30

**Decision:**

Accept (spotlight)

**Comment:**

This paper proposes MORetro*: a retrosynthesis algorithm that generates an entire Pareto front of synthesis routes, which prioritize different trade-offs between objectives such as cost, toxicity, or yield. Authors analyse the approach theoretically, and then present experimental evidence, showing it produces high-quality Pareto fronts, catching solutions that single-objective optimization algorithms may overlook.

On the positive side, reviewers considered the work to be elegant, robust, and well-executed. They mentioned it aligns well with real-world use, and follows incentives within pharma that encourage exploring a diverse suite of potential synthesis pathways. Reviewers also found the paper to be clear and well-written, and praised the strong rebuttal, which provided several additional results, and added useful discussion on limitations and future directions.

On the negative side, some reviewers noted a minor degradation in solve rate when compared to Retro*. They highlighted that MORetro* is currently limited to additive scores that decompose as a sum over reactions in the route, and that the current evaluation uses three objectives, with unclear scaling to many more than this. Authors acknowledged both limitations, adding appropriate discussion into the paper, and framing them as future work. Some reviewers also mentioned the work is not a fundamental ML breakthrough, and rather a (well-executed) integration of several existing algorithms; this however does not preclude acceptance, especially in the AI for Science domain. Finally, reviewers also mentioned the objectives evaluated in this paper are approximated with proxies, and thus the exact results should be taken with a grain of salt. Authors argued that their work tries to demonstrate that optimization towards a set of objectives is feasible in general, and, according to reviewers, it does achieve that goal sufficiently well.

In aggregate, all four reviewers were in favour of accepting this paper; although they noted room for improvement across several axes, they considered the current state of the manuscript to already be a strong self-contained publication-ready piece of work.

After reviewing all discussion, and reading the paper myself, I decided to lean into the reviewer consensus, and also recommend acceptance.